# Dietary Glycemic Index and Load and the Risk of Type 2 Diabetes: A Systematic Review and Updated Meta-Analyses of Prospective Cohort Studies

**DOI:** 10.3390/nu11061280

**Published:** 2019-06-05

**Authors:** Geoffrey Livesey, Richard Taylor, Helen F. Livesey, Anette E. Buyken, David J. A. Jenkins, Livia S. A. Augustin, John L. Sievenpiper, Alan W. Barclay, Simin Liu, Thomas M. S. Wolever, Walter C. Willett, Furio Brighenti, Jordi Salas-Salvadó, Inger Björck, Salwa W. Rizkalla, Gabriele Riccardi, Carlo La Vecchia, Antonio Ceriello, Antonia Trichopoulou, Andrea Poli, Arne Astrup, Cyril W. C. Kendall, Marie-Ann Ha, Sara Baer-Sinnott, Jennie C. Brand-Miller

**Affiliations:** 1Independent Nutrition Logic Ltd, 21 Bellrope Lane, Wymondham NR180QX, UK; richard.mi.taylor1@gmail.com (R.T.); helenlivesey@inlogic.co.uk (H.F.L.); 2Institute of Nutrition, Consumption and Health, Faculty of Natural Sciences, Paderborn University, 33098 Paderborn, Germany; anette.buyken@uni-paderborn.de; 3Departments of Nutritional Science and Medicine, Faculty of Medicine, University of Toronto, Toronto, ON M5S 1A8, Canada; david.jenkins@utoronto.ca (D.J.A.J.); john.sievenpiper@alumni.utoronto.ca (J.L.S.); thomas.wolever@utoronto.ca (T.M.S.W.); cyril.kendall@utoronto.ca (C.W.C.K.); 4Clinical Nutrition and Risk Factor Modification Centre, St. Michael’s Hospital, Toronto, ON M5C 2T2, Canada; livia.augustin@utoronto.ca; 5Division of Endocrinology & Metabolism, Department of Medicine, St. Michael’s Hospital, Toronto, ON M5C 2T2, Canada; 6Li Ka Shing Knowledge Institute, St. Michael’s Hospital, Toronto, ON M5C 2T2, Canada; 7Epidemiology, Istituto Nazionale Tumori IRCCS “Fondazione G. Pascale”, 80131 Napoli, Italy; 8Glycemic Index Foundation, 26 Arundel St, Glebe, NSW 2037 Sydney, Australia; alanb@gifoundation.org.au; 9Department of Epidemiology and Medicine, Brown University, Providence, RI 02912, USA; simin_liu@brown.edu; 10Departments of Nutrition and Epidemiology, Harvard T. H. Chan School of Public Health and Harvard Medical School, Boston, MA 02115, USA; wwillett@hsph.harvard.edu; 11Department of Food and Drug, University of Parma, 43120 Parma, Italy; furio.brighenti@unipr.it; 12Human Nutrition Unit, Department of Biochemistry and Biotechnology, Faculty of Medicine and Health Sciences, Institut d’Investigació Sanitària Pere Virgili (IISPV), Rovira i Virgili University, 43201 Reus, Spain; jordi.salas@urv.cat; 13Fisiopatología de la Obesidad y Nutrición (CIBEROBN), Instituto de Salud Carlos III, 27400 Madrid, Spain; 14Retired from Food for Health Science Centre, Antidiabetic Food Centre, Lund University, S-221 00 Lund, Sweden; inger@innovafood.se; 15Institute of Cardiometabolism and Nutrition, ICAN, Pitié Salpêtrière Hospital, F75013 Paris, France; salwa.rizkalla3@gmail.com; 16Department of Clinical Medicine and Surgery, Federico II University, 80147 Naples, Italy; riccardi@unina.it; 17Department of Clinical Sciences and Community Health, Università degli Studi di Milano, 201330 Milan, Italy; carlo.lavecchia@unimi.it; 18IRCCS MultiMedica, Diabetes Department, Sesto San Giovanni, 20099 Milan, Italy; ACERIELL@clinic.cat; 19Hellenic Health Foundation, Alexandroupoleos 23, 11527 Athens, Greece; atrichopoulou@hhf-greece.gr; 20Nutrition Foundation of Italy, Viale Tunisia 38, I-20124 Milan, Italy; poli@nutrition-foundation.it; 21Department of Nutrition, Exercise and Sports (NEXS) Faculty of Science, University of Copenhagen, 2200 Copenhagen, Denmark; ast@nexs.ku.dk; 22College of Pharmacy and Nutrition, University of Saskatchewan, Saskatoon, SK S7N 5B5, Canada; 23Spinney Nutrition, Shirwell, Barnstaple, Devon EX31 4JR, UK; nutrition@thespinney.co.uk; 24Oldways, Boston, MA 02116, USA; sara@oldwayspt.org; 25Charles Perkins Centre and School of Life and Environmental Sciences, University of Sydney, Sydney NSW 2006, Australia; jennie.brandmiller@sydney.edu.au

**Keywords:** glycemic index, glycemic load, dietary fiber, protein, alcohol, type 2 diabetes, cohort studies, epidemiology, meta-analysis

## Abstract

Published meta-analyses indicate significant but inconsistent incident type-2 diabetes (T2D)-dietary glycemic index (GI) and glycemic load (GL) risk ratios or risk relations (RR). It is now over a decade ago that a published meta-analysis used a predefined standard to identify valid studies. Considering valid studies only, and using random effects dose–response meta-analysis (DRM) while withdrawing spurious results (*p* < 0.05), we ascertained whether these relations would support nutrition guidance, specifically for an RR > 1.20 with a lower 95% confidence limit >1.10 across typical intakes (approximately 10th to 90th percentiles of population intakes). The combined T2D–GI RR was 1.27 (1.15–1.40) (*p* < 0.001, *n* = 10 studies) per 10 units GI, while that for the T2D–GL RR was 1.26 (1.15–1.37) (*p* < 0.001, *n* = 15) per 80 g/d GL in a 2000 kcal (8400 kJ) diet. The corresponding global DRM using restricted cubic splines were 1.87 (1.56–2.25) (*p* < 0.001, *n* = 10) and 1.89 (1.66–2.16) (*p* < 0.001, *n* = 15) from 47.6 to 76.1 units GI and 73 to 257 g/d GL in a 2000 kcal diet, respectively. In conclusion, among adults initially in good health, diets higher in GI or GL were robustly associated with incident T2D. Together with mechanistic and other data, this supports that consideration should be given to these dietary risk factors in nutrition advice. Concerning the public health relevance at the global level, our evidence indicates that GI and GL are substantial food markers predicting the development of T2D worldwide, for persons of European ancestry and of East Asian ancestry.

## 1. Introduction

According to the International Diabetes Federation, type 2 diabetes (T2D) is a huge and growing problem and costs to society are high and escalating [1]. Reducing the burden of diabetes by prevention is a major goal. Potential modifiable lifestyle choices that affect a person’s risk include being overweight or obese (particularly central obesity), smoking, low physical activity, and the consumption of diets rich in refined grains and alcohol, and low in dietary fiber and whole grains. Diets with a high glycemic index (GI) or load (GL) have also been reported to increase the risk of this metabolic condition [2]. (See endnote ***a*** after the Discussion for definitions of the glycemic index and glycemic load.)

Although T2D is a disease of abnormal carbohydrate metabolism, meta-analyses of results from prospective cohort studies have shown no clear association between the sum of all digestible carbohydrates ingested and incident T2D [3,4,5]. Likewise, no clear association is evident between the levels of starch [6] and sucrose [7] intake and T2D. However, added sugars have been reported to increase the risk of T2D when taken in a liquid form, such as soft drinks and fruit juice drinks—potentially due to an overconsumption of energy contributing to obesity [8].

In contrast to meta-analyses on carbohydrates, starch, and sucrose, published meta-analyses over the last decade have shown significant, but inconsistent associations between incident T2D and GI [5,9,10,11,12] and GL [5,10,11,13,14]. Outcomes published include T2D–GI RR values that ranged from 1.12 to 1.45, and T2D–GL RR values that also ranged from 1.12 to 1.45. For these outcomes, exposures to the GI and GL approximately spanned from the 10th to the 90th percentiles for typical population intakes of GI and GL. Moreover, a recent high profile publication with meta-analyses has indicated either 1.05 or 1.12 for GI and zero for GL, but it is unreliable for multiple reasons [15] and has been criticized for its viewpoint on the GI [16]. This variability in the strength of association among the published T2D–GI and GL meta-analyses is unhelpful since for inclusion within nutrition guidance, it has been recommended that relative risks from prospective cohort studies should fall to <0.83 (with a 95% upper confidence limit < 0.91) or rise to >1.20 (with a lower confidence limit > 1.10) across population quintiles (i.e., from the 10th to the 90th percentile of population intakes under ideal circumstances) [15,17].

What is also unhelpful is that most studies on these relations have used extreme quantile meta-analyses (EQMs) combining studies with different definitions of exposure, across tertiles, quartiles, and mostly quintiles, so the strength of association across 80 percentiles (10th to 90th) may be a source of underestimation. The 80th percentile range corresponds approximately to intakes from the 1st to 5th quintiles.

Among all the published meta-analyses associating incident T2D with GI and GL [5,9,10,11,12,13], only one used a well-performed quantitative dose-response meta-analysis (DRM) for GI [5], and only two did so for GL [5,13]. The others used EQMs, which use only a fraction of the available information. Further, only one meta-analysis [9] restricted their study selection to those applying truly valid dietary instruments, which were for nutrient amounts in general (validity correlation > 0.5) [18]. For GI and GL, a validity correlation for carbohydrate > 0.55 is considered appropriate when assessing coronary heart disease and GI and GL risk relations [19], and was applied here. This higher value (0.55) was selected based on the study of Brunner et al. [18], who recommended, in general, a value of 0.5 for nutrients, which was adopted by Barclay et al. in 2008 [9] in their studies of T2D GI and GL risk ratios, to which 10% was added to allow for any error of the estimation.

Since the first and only meta-analysis of studies utilizing the T2D–GI and GL risk ratios with truly valid dietary instruments, which was published in 2008 [9], 11.6 million person-years of additional information has become available. Therefore, we updated the meta-analyses (as in [9]) for both GI and GL, again using studies with truly valid dietary instruments. A difference is that we defined a truly valid dietary instrument as one that has a validity correlation > 0.55 for carbohydrates [19] rather than 0.5 for nutrients in general. A further difference is that we used preferred quantitative dose–response meta-analysis (DRM) [5,19,20,21].

Dose response meta-analysis (DRM) is preferred to extreme quantile meta-analysis (EQM) because it uses more of the available data and allows an assessment of the risk ratios or risk relations (RR) per unit increment of exposure, which is particularly helpful when undertaking global DRM (i.e., increments in RR across all levels of exposure from lowest to highest exposures worldwide) and when looking for potential nonlinear associations in the dose-response over the global range [5,13,19]. A drawback is that the use of dietary instruments that do not collect representative information leads to an overdispersion of the range of intake estimates, which has a major consequence—an underestimation of the dose-response relationships, suggested to be up to a 75% reduction [22]. Further error arises when undertaking dose-response meta-analysis and is two-fold. First, overdispersion lowers the risk relation and its standard error. Second, the overdispersion increases the weight of the study in meta-analysis. For example; a doubling of the dispersion compared to the real dispersion halves both the risk relation and its standard error but doubles the contribution the study RR makes to the weighted mean RR in fixed effects meta-analysis (and less than doubles the weighting in random effects meta-analysis). In other words, poor dietary instruments overweigh studies, thus reporting lower relative risks than in reality. Other problems encountered in meta-analyses involving GI and GL have been described elsewhere [15,19]. Meta-analyses of observational studies should be done with due care [23,24]. This includes minimizing problems by excluding studies applying inferior (invalid) dietary instruments [9,19] or retaining them only when a validity correlation is used as a covariate [13].

Therefore, the primary aim of this paper was to comprehensively update the evidence using a systematic review and meta-analyses of the relationships between GI, GL, and T2D from prospective cohort studies using truly valid dietary instruments. Participants were adults in good health at baseline, representing the general population from any country worldwide. Comparisons were of a higher versus lower exposure of GI and GL in dose-response meta-analyses. Outcomes were the rates of change in relative risks (i.e., risk relations) with increasingly higher GI and GL diets. A secondary aim was to address recent hypotheses for inconsistencies in the results among studies (Section 2.2.1), which has not yet been examined when initially selecting studies that have truly valid dietary instruments, and is an important aspect of a systematic review and meta-analysis of observational studies [24]. A further aim was to assess these relations over the global scale since the range of GI and GL intakes are greater worldwide (globally) than within national jurisdictions (locally) [5,13,19]. In a second paper [15], we examined whether the association of GI and GL with the development of T2D might be causal through the application of the Bradford–Hill criteria [25].

## 2. Methods

We followed both the Preferred Reporting Items for Systematic Reviews and Meta-Analyses or PRISMA protocol [26] and the meta-analysis of observational studies in epidemiology (MOOSE) proposals for reporting [27].

### 2.1. Systematic Review

#### 2.1.1. Literature Sources and Searches

Searches were conducted on MEDLINE and EMBASE using PROQUEST via the Royal Society of Medicine (RSM), London, UK from 1946 to 6 December 2018, and all dates to 6 December 2018 via the Cochrane Library of Systematic Reviews and the PROSPERO Register of Systematic Reviews. The literature search strategy (see Appendix A) aimed for relevant peer reviewed prospective cohort studies on incident T2D related to GI or GL and was developed in collaboration with LitSearch expertise at the RSM. Manual searches were also made of retrieved peer reviewed publications entering the review process and of prior published meta-analyses.

#### 2.1.2. Inclusions

Studies had to be prospective cohort studies, which provide the highest level of evidence among epidemiological studies, relevant to adult public health investigations of the association between incident T2D and GI and/or GL. Studies with at least two defined categories of exposure (by quantiles, quantities, and/or range, such as the standard deviation of intakes), reporting estimates of relative risk (risk ratio, hazard ratio, odds ratios), measures of uncertainty (standard errors or confidence intervals), and controlling (adjusted) for confounding factors. Studies with a follow-up duration ≥ 4 years were included. Ascertainments of T2D by self-reporting and by clinical assessment were included. Studies of adults of any age, sex, ethnicity, from any region worldwide, and published at any time were included without language restriction.

#### 2.1.3. Exclusions

Exclusion criteria were: Studies that were not observational, longitudinal, and prospective; studies not reporting the exclusion of persons with T2D at baseline; studies including a high proportion of persons with glucose intolerance or ill health; studies not fully reporting on their fully adjusted model; and studies not reporting validation results for their dietary instrument for carbohydrate intake either directly or by citation of a validation study. The Appendix A provides specific details of why certain studies [3,10,28,29,30,31,32,33] did not meet our inclusion or exclusion criteria.

#### 2.1.4. Selection

For the primary outcomes, studies had to apply dietary instruments with validity correlations for carbohydrate > 0.55. Studies that proved to be significant outliers (Section 2.1.7) were excluded, but retained for inclusion whenever a meta-analytical model used proved them as inlying for secondary outcomes. Likewise, studies excluded from the primary meta-analysis because they had inferior dietary instruments (validity correlations for carbohydrate ≤ 0.55) were similarly retained and used where the validity correlation was a covariate.

For the prospective cohort studies in peer reviewed publications that were reported in more than one publication, we used observations for the longest follow-up duration. If the results from the two longest durations within a study were inconsistent, we precombined them by random effects meta-regression and calculated the output standard error as (τ^2^ + Ʃσ_i_^2^/n)^0.5^, where τ^2^ is the heterogeneity, σ_i_ is the variance within the ith study, and n is the number of studies combined (2 in this situation). This ensured the study was represented only once while representing the higher level of uncertainty due to the inconsistency between reports in the subsequent meta-analyses. The sensitivity of our primary outcomes to this procedure was examined (Appendix A).

#### 2.1.5. Data Extraction and Calculation

Data were extracted by two of three independent reviewers (GL, HFL, RT), and disparities were resolved by agreement.

GI and GL by quantiles (means or medians or calculated as shown in the footnotes to Appendix A).The reference standard used (glucose or white bread)—so to adjust to a common reference standard of glucose (GI of white bread = 100 on the bread scale, and 70 on the glucose scale).Relative risks for T2D (risk ratios, hazard ratios or odds ratios) and their 95% confidence intervals (95% CI) by quantiles or by risk relations (increment in RR per units of increased exposure).Energy intake by quantiles and the intake of energy to which GL was adjusted.Whether RR values from the prospective cohort studies were adjusted for protein intake simultaneously with adjustments for energy, fats, and fiber intakes (See endnote ***b*** after the Discussion for a hypothesis of when, in original studies, adjusting for protein intake might make an adjustment for carbohydrate intake, and why it might not).Whether study-level adjustments were made for fiber, magnesium, red meat, macronutrients, and family history of diabetes (FHD).Study population’s average alcohol (ALC) consumption and fiber (including type) intakes.Sex, as a fraction of the men in the sampled population (SEX).Ethnicity (ETH).Population region.The person-years of follow-up, the number of cases, and the number of persons followed up.Duration of follow-up in years (FUY).Whether the dietary instruments used were validated and the level of validity for carbohydrate as indicated by the validity correlation coefficient for carbohydrate (CORR).The number of repeated uses of the dietary instrument over the duration of the follow-up.The method used for the ascertainment of T2D (self-report or clinical report).Body mass index (BMI) values (kg/m^2^) for subcohorts differing by BMI.Adverse events of any kind reported in the original studies.Whether the authors of the prospective cohort studies were potentially conflicted by sources of funding.

Missing information was either calculated where possible (Appendix A) or obtained from corresponding published dietary instrument validation studies or sought by correspondence with the authors.

#### 2.1.6. Study Quality

For individual prospective cohort studies, quality was assessed using the Newcastle–Ottawa score (NOS) for observational studies [34,35]. The score was modified to reduce comparability by 1 star for studies using dietary instruments with energy-adjusted deattenuated correlation coefficients for carbohydrate ≤ 0.55. A copy of the procedure is included in the Appendix A together with the guidance used.

#### 2.1.7. Additional Analyses

Because epidemiological studies have the potential to generate precise, but spurious results, the possibility of an outlier study result was examined by meta-regression using an indicator variable for a suspected study (one for which the study confidence interval was not overlapping that for the combined studies’ RR), which assumed normal distribution [36]. Discrimination was set at *p* < 0.05 without adjustment for multiple comparisons. Outlier studies were withdrawn, but reserved for re-inclusion with any new analytical model.

Sensitivity analyses were undertaken, where specified, by dropping individual studies in turn, by dropping significant groups of studies (e.g., health professionals), by re-inclusion of outlier studies, by exchange of results from the same study at different durations of follow-up, and by adding studies for which information was incomplete using assumptions for missing information, as specified in the Appendix A.

Small-study effects were assessed from funnel plot asymmetry (e.g., publication bias, study selection bias) using the Duval and Tweedie nonparametric trim-and-fill method (‘metatrim’ version 1.0.3 in Stata), and by a Galbraith-like regression (‘regress’ version 1.2.7 in STATA) of logRR_i_ •N_i_ on N_i_, where RR_i_ is the study-level relative risk and N_i_ is the study-level population sample size (number of persons followed up) [37,38].

Subgroup meta-analysis with or without covariates was used to investigate any potential dependence of the risk relations on the characteristics reported by categories, e.g., participants’ SEX and BMI (body mass index (kg/m^2^)). Meta-regression was used to adjust the RR for covariates and to examine the hypothesized covariates and the significance of difference between subgroups.

### 2.2. Data Synthesis and Meta-Analysis

#### 2.2.1. Hypotheses and Statistics

Individual study-level summary measures of the relative risks for the most fully study-level-adjusted models were used, though not including adjustments for C-reactive protein as occurred in one publication. The primary outcomes were combined risk relations (RR) from those studies using dietary instruments having a validity correlation for carbohydrate (CORR) > 0.55 [9,13,18]. Those studies with values ≤0.55 were retained for use in analytical models applying CORR as a covariate in secondary outcomes. Outlier studies (*p* < 0.05, Section 2.1.7) were withdrawn owing to the occurrence of spurious results in epidemiology generally, which can be of high precision [39]; these were also retained for use in accommodative meta-analytical models addressing secondary outcomes in which they became inlying.

Secondary outcomes concerned the significance of potential sources of heterogeneity (τ^2^) or inconsistency (I^2^) when excluding studies with validity correlations < 0.55 for carbohydrate unless specified differently, as in *i* below, or when analytical models accommodated the validity correlation as a covariate:(i)Confirmation of low or lower association among studies applying a dietary instrument with a validity correlation < 0.55 for carbohydrate [9,13].(ii)Ascertainment of T2D by self-report and or by clinical record [9].(iii)Study-level adjustment for protein intake simultaneously with energy and other macronutrient intakes except carbohydrate (see endnote ***b*** after the Discussion for a hypothesis of when an adjustment for carbohydrate may have been made and why it may have not) [10].(iv)Population’s average ALC consumption [40].(v)Sex of participants, including whether results for mixed sexes could be due to the inclusion of men [13,41,42].(vi)Duration of follow-up [5,13].(vii)Number of dietary assessments [10].(viii)Body mass index [43,44,45,46].(ix)Specific nutrient adjustments at the study level (a posteriori).(x)Study-level adjustment for a family history of diabetes (FHD) [5,45].(xi)Study size, with large studies generally being considered important in longitudinal epidemiology, notable ones in general averaging 60,000 to 70,000 participants or more [22], and contrast intervention trials where small studies commonly appear as biased towards larger treatment effects [47,48].(xii)Geographical region (a posteriori).

As referenced in (i) to (xii), hypothesized sources of heterogeneity and inconsistency were suspected a priori unless specified differently. Additionally, the possibility of studies reporting high T2D–GI or GL RR values due to the failure of making study-level adjustments for fiber and other nutrients was also addressed.

A two-step meta-analysis procedure was used for dose-response meta-analysis (DRM). Step 1 used the generalized least-squares method for trend estimation of the summarized dose-response data (*glst v9.2*) [20,21] on individual studies, which provided individual study dose-response RR values. Step 2 then combined the individual study mean dose-response RR values by meta-analysis without covariates (*metan v3.03*) or with covariates by meta-regression (*metareg v6.1*) using the method of moments and random effects [49]. Meta-analysis without covariates used both the fixed and random effects of Mantel and Haenszel [50] and DerSimonian and Laird [51], respectively. The random effects method was used for inconsistent results (I^2^ > 0%), which resolved fixed effects when results were consistent (I^2^ = 0).

In the tables, the fixed effects results are shown in addition to the random effects results, in which there is discussion applicable to intervention studies [49,52] (I^2^ = 0); this is because some users have considered fixed effects more appropriate when the range of observational study sizes is large [52] when random effects is thought to under-weight large studies. Thus, a fixed effect was applied alone to one published meta-analysis of T2D-GI and GL relative risks [10]. However, random effects have a more generally accepted theoretical basis, particularly for observational studies for which study results are typically quite variable. Our primary and secondary outcomes were random effects.

Dose was applied as increments in GI or GL intakes. For quantitative DRM, T2D–GI and GL risk relations were expressed as RR per 10 units GI and 80 g/d GL adjusted to a 2000 kcal (8400 kJ) diet, respectively. These rates corresponded approximately to an increment in the RR per average dose increment from the 10th to the 90th percentiles of population intakes for eligible studies (CORR > 0.55). This was the principal method used throughout this review. Extreme quantile meta-analysis (EQM) was used where quantitative information on GI or GL was not available, which was the case for BMI subgroups at the study level in the original prospective cohort studies.

All statistical analyses were undertaken in Stata software (release 11.2 SE, 2009; StataCorp LP College Station, TX, USA) using the “*mais*” installation [53].

#### 2.2.2. Global Dose-Response Meta-Analysis

This was undertaken using the generalized least-squares method for trend estimation of the summarized dose-response data (*glst v90*) in the one step pool first modality [20,54] repeated as described previously [19]. The method introduced a third error term (σ_f_), which was due to forecasting the graphical location of each of the second to last studies entered, and which was additive with τ^2^ + σ^2^ the heterogeneity among studies (τ^2^) and the variance within studies (σ^2^). Inconsistency (I^2^%) was calculated both as usual, 100•τ^2^/(τ^2^ + σ^2^), and as 100•τ^2^/(τ^2^ + σ^2^ + σ_f_^2^) to assess the extent to which σ_f_ contributed to any reduction of inconsistency among studies by allowing a nonlinear dose-response. To allow a nonlinear response, all exposure measures in all studies included in the analysis were used to generate restricted cubic spine determinants with knots at the 10th, 50th, and 90th percentile of exposure. Random effects analysis was used. Confidence limits placed in the text were calculated using ±standard errors (SE) of (τ^2^ + σ^2^ + σ_f_^2^)^0.5^ while graphs show the confidence limit calculated using both (τ^2^ + σ^2^)^0.5^ and (τ^2^ + σ^2^ + σ_f_^2^)^0.5^ to visualize how much wider the limits fell due to forecasting the graphical location of added studies relative to the first included study [19].

#### 2.2.3. Statistical Tests

The z test was used for combined means, covariates, and outliers; the t test for small-study effects; and the χ^2^ test for heterogeneity (τ^2^) or inconsistency (I^2^). τ^2^ and I^2^ can be interpreted only approximately in meta-analysis with or without covariates [55,56]. The possibility of a false significance in a multiple covariate was tested by the method of permutations (*permute* (10,000) in Stata).

#### 2.2.4. Data and Terminology

Meta-analyses used log RR, which were exponentiated to the unlogged form for presentation (unless specified otherwise). Exposures were the glycemic index (GI, % on the glucose standard) and the glycemic load (GL = GI × amount of carbohydrate (g)/100). The RR across 10 units of GI was expressed as RR per 10 units GI. The RR across 80 g/d GL was expressed as 80 g/d GL per 2000 kcal (8400 kJ) diet. These corresponded to the average range of GI and GL values from the 10th to the 90th percentile of population samples eligible in the present study (10.2 units GI and 80.9 g/d GL per 2000 kcal (8400 kJ) diet.

#### 2.2.5. Presentation

The T2D–GI and GL risk relations are presented in the tables and/or figures by both fixed and random effects models. For brevity, the text refers only to the random effects results, which became a fixed effect for instances where I^2^ = 0.

## 3. Results

### 3.1. The Literature Search

The literature search (Section 2.1.1) identified a total of 801 records, 781 from a simultaneous search of MEDLINE and EMBASE, 19 from PROSPERO and the Cochrane Library, and 1 additional study from our prior meta-analysis [13] (Figure 1). After the removal of duplicate and irrelevant records based on an examination of the titles and abstracts, 34 records were retrieved for further evaluation (Figure 1). This identified 10 reports that did not meet the inclusion/exclusion criteria; two were not original studies, [28,57], one was not of a prospective design [32], four used an ineligible population [30,31,58,59], two did not address the T2D–GI or GL risk relation [3,29], and one did not use a dietary questionnaire including validation for carbohydrate or carbohydrate foods [33] (see Appendix A, Section 3, paragraphs a–e for further detail). This left 24 reports that were further evaluated separately for the T2D–GI risk relation in Section 3.2 and the T2D–GL risk relation in Section 3.3.

The earliest study identified was published in 1997 [60]. Prior to this date, only two potentially relevant records were retrieved, but neither was a relevant study.

### 3.2. The T2D–GI Risk Relation

We sought prospective cohort studies on the T2D–GI or GL relative risk (point estimates) or risk relations (rate estimates) that used truly valid dietary instruments (CORR > 0.55, see Methods Section 2.1.6) or could be included when CORR was used as a covariate. From the 24 published reports (Figure 1 and Figure 2) on the T2D–GI and T2D–GL risk relations, three did not report on the T2D–GI risk relation [4,42,61] and five did not include the longest duration of follow-up within a study (with two from one publication addressing NHS II and HPFS) [10,60,62,63] (Figure 2). For further details, see Section 3 (f and g) of the Appendix A. The remaining 16 reports included 18 studies with enough available information to perform dose-response meta-analysis (DRM) (Figure 2). Among these 18 studies, 7 had dietary instrument correlation coefficients that were invalid by our criterion (CORR ≤ 0.55) (listed in Table 1, footnote *b*), leaving 11 truly valid studies of which one proved to be a significant outlier (Figure 2 and Section 3.2.2), which was withdrawn to only be re-included should it fit an arising circumstance.

#### 3.2.1. Study Characteristics

The individual study characteristics are given in Appendix A. For the 10 studies using valid dietary instruments (validity correlations > 0.55) that were available for DRM and were not outliers: Countries or regions included were the USA, Japan, Europe, China, and Australia; sexes included men-only studies (n = 3), women-only studies (n = 3), and mixed-sex studies (n = 4); baseline ages (mean per study) ranged from 36 to 75 years. All studies excluded T2D at baseline, including one study that was not included in a priori meta-analysis [71]; and dietary exposures in most studies were ascertained using food frequency questionnaires (FFQs)—one used a diet history questionnaire [71] and one used an undefined questionnaire plus a structured interview with a trained dietician [72]. The dietary instruments had variable quality with respect to the validity correlation for carbohydrate, which ranged from 0.56 to 0.80. Compared with the large studies in epidemiology averaging 60,000 to 70,000 participants prior to 1996 [22], of those investigations on the T2D–GI risk relation, three studies were of this size, one from China [46] and two from the USA [10,40]. The remaining studies used 2000 to 40,000 participants. The studies were adjusted for major potential confounders (see Section 3.2.12 and Section 3.3.7). The average BMI values of the studies ranged from 23 to 27 kg/m^2^.

Individual study quality, assessed on the Newcastle–Ottawa score (NOS), ranged from 5 to 8 for possible scores of 0 to 9. Those studies with CORR ≤ 0.55 had a mean NOS of 5.7 (SD 0.96) whereas studies with CORR > 0.55 has a score of 7.4 (SD 0.7). From the outset, there was little obvious risk of bias other than for the dietary instrument validity. Studies reported no adverse events except the outcomes of focus and none of the studies reported any industry funding.

#### 3.2.2. Combined Observations

Extracted study-level data and related calculated values are given in Appendix A. The random effects combined studies’ T2D–GI risk relation obtained by DRM was approximately 5 to 6 times higher for those studies using a truly valid dietary instrument than those that did not. Without this validity, the random effects risk relation was nonsignificant at 1.05 (0.97–1.14) (*p* = 0.271, n = 7) per 10 units GI compared with the significant relation of 1.27 (1.15–1.40) (*p* < 0.001, n = 10) per 10 units GI when valid (Table 1 and Figure 3). As assessed by random effects meta-regression, these two subgroups were significantly different (*p* = 0.004). Heterogeneity was high (Table 1) and was attributable (see Section 3.2.13).

#### 3.2.3. Clinical Versus Self-Report of Diabetes

Keeping only those studies with valid dietary instruments (Section 3.2.2) and valid ascertainment of T2D (i.e., clinically reported, which was estimated as self-reported in ≤3% of the population sample), the random effects RR was 1.40 (1.29–1.51) (*p* < 0.001, n = 6) per 10 units GI with nonsignificant inconsistency (I^2^ = 14%) and heterogeneity (τ^2^ = 0.0015) (not tabulated). The included studies were: Sakurai et al. (2012) [71], Sahyoun et al. (2008) [70], Barclay et al. (2007) [68], Bhupathiraju et al. (2014) (HPS) [10], Mekary et al. (2011) (NHS I) [40], and Bhupathiraju et al. (2014) (NHS II) [10]. The studies of Simlia et al. [40] and van Woudenbergh et al. [72] were significant outliers (*p* < 0.001 and *p* = 0.029, respectively).

Again, keeping only those studies with validity correlations > 0.55 (Section 3.2.2), we assessed how clinical and self-reported T2D–GI RR values differed using meta-regression. For this, clinically reported, self-reported, and a mix of self and clinically reported studies were represented as a continuous variable zero centered on clinically reported studies (see Appendix A, footnote *a*). Self-reported RR values were lower by 0.07 (–0.08–0.25), but not significantly (*p* = 0.48, n = 11) while the study of Simila [41] was an outlier and low (*p* = 0.003). In this model, the RR was 1.35 (1.22–1.49) (*p* < 0.001, n = 11), consistent with the higher RR found when selecting only those studies using clinically reported T2D in 97% or more cases.

#### 3.2.4. Funnel, Trim-and-Fill, and Galbraith-Like Plot Asymmetries

A funnel plot of the T2D–GI risk relations for studies with dietary instrument validity correlations >0.55 for carbohydrate and no outliers showed no evidence of asymmetry (Figure 4). Trim-and-fill analysis indicated no evidence of missing studies (no filling with hypothetical missing observations) and so the unadjusted combined random effects dose-response T2D–GI risk relation was 1.27 (1.15–1.40) (*p* < 0.001, n = 10) (τ^2^ = 0.013, *p* < 0.001). These outcomes were identical to those for the corresponding parametric meta-analysis (Table 1 panel 2, n = 10, random effects). Galbraith-like analysis indicated asymmetry was nonsignificant (Egger test, *p* = 0.681, n = 10).

#### 3.2.5. Sensitivity to Individual Studies

Sensitivity analysis was conducted by removing one study at a time for our primary outcome, for which the T2D–GI risk relation was 1.27 (1.15–1.40) (Table 1 panel 2, n = 10, random effects). The highest RR occurred after dropping the small study (4366 persons) of van Woudenbergh et al. (2011) [72], after which the random effect RR became 1.29 (1.17–1.43). The lowest RR occurred after dropping the large study (75,457 persons) of Mekary et al. (2011) [40], after which the random effects RR became 1.22 (1.13–1.32). All these values therefore met the criteria of interest for public health (RR values > 1.20 with lower 95% CL > 1.10.)

#### 3.2.6. Mixed-Sex Studies

In this study, we addressed concern arising in the literature [41] that the T2D–GI risk relation may be lower in men than in women, which has arisen because mixed-sex studies have sometimes yielded low T2D–GI risk relations. We assessed whether other explanations might exist to explain the low T2D–GI risk relations in mixed-sex studies undertaken to date.

Eight mixed-sex studies from seven publications [65,66,67,68,69,72,73] were available with information sufficient for dose-response meta-analysis. Combined, these yielded a fixed and random effects T2D–GI risk relation of only 1.06 (0.99–1.14) (*p* = 0.07, n = 8) per 10 units GI (I^2^ = 0%, τ^2^ < 0.0001, *p* = 0.559) (data not tabulated). This combined RR value was far below the expected values for the dilution of observations on women by the observations on men (Section 3.2.7). Other explanations are available. Of the eight mixed-sex studies, only four avoided the use of the self-reporting of diabetes [65,68,70,72], only four had used dietary instruments with correlation coefficients for carbohydrate > 0.55 [68,69,70,72], and only two avoided both the self-reporting of diabetes and had validity correlations for carbohydrate > 0.55 [68,72].

Thus, a high proportion of the mixed-sex studies that reported low–T2D-GI relations were due to reasons other than the inclusion of men within the study.

#### 3.2.7. Single-Sex Studies

Among the 10 studies that used a valid dietary instrument (Section 3.2.2), 6 were conducted on women-only or men-only (Table 1). The random effects combined mean T2D–GI risk relation for these single-sex studies was 1.29 (1.15–1.45) per 10 units GI.

The six studies were divided evenly into subgroups of three women-only and three men-only studies. In women, the random effects RR was 1.29 (1.10–1.50). A similarly high T2D–GI risk relation was found for men with random effects of 1.31 (1.06–1.63). In these analyses, the RR values were significant (*p* < 0.02 in one instance and *p* < 0.001 in all others) (Table 1).

The T2D–GI risk relation for men-only and women-only studies combined was >1.20 with a lower confidence interval > 1.10, with no difference between men and women.

#### 3.2.8. Duration of Follow-Up

A prior dose-response meta-analysis of the T2D–GI risk relation indicated the risk may increase with the duration of follow-up in years (FUY) [5]. Combined with the finding that the T2D–glycemic load risk relation was not related to the duration of follow-up during multi-covariate meta-analysis [13], this led us to address this hypothesis at present for the T2D–GI risk relation. We confirmed the observation in [5] (*p* < 0.001). However, among-study observations should be regarded with caution.

More reliable evidence should come from analysis within studies. Three prospective cohort studies (NHS I, NHS II and HPFS) had results for more than one duration of follow-up. These followed up from baseline to 6 and 26 years, baseline to 8 and 18 years, and baseline to 6 and 22 years, respectively [10,45,60,74]. Repeated administration of the dietary instrument across the duration of follow-up was used to account for any changes in exposure with time from baseline. A random effects meta-regression for follow-up years as a covariate and the three constants, one for each study, suggested a nonsignificant (*p* = 0.30) increment in the relative risk of +0.05 (–0.04–0.16) per 10 units GI per 10 years.

Although we may conclude from the within study analysis that the T2D–GI risk relation did not increase significantly with the duration of the study, it did not decrease significantly (or at all) with the duration of follow-up.

#### 3.2.9. Number of Dietary Assessments

A positive association between the T2D–GI risk relation and the number of dietary assessments among prospective cohort studies has been suggested [10], but neither prior dose-response analysis nor consideration of the validity of the dietary instruments used has been undertaken.

Among the studies, we found dose-response RR values to be associated with the number of dietary assessments when selecting only those studies using dietary instruments with validity correlations for carbohydrate > 0.55. The RR was 1.04 (1.02–1.06) (*p* < 0.001, n = 10) per 10 units GI per dietary assessment (I^2^ = 8%, τ^2^ = 0.0008, *p* = 0.37). In this analysis, the study of Simila et al. was an outlier (*p* = 0.040) and was excluded. This was based on the resu1ts combined from (i) four studies using a single (baseline) dietary assessment [9,69,71,72], (ii) one study using two dietary assessments [46], (iii) one study using three dietary assessments [44], (iv) two studies using six dietary assessments [10], and (v) one study using seven dietary assessments [40]. However, the size of this increment should be considered with caution because it arose from the among-study RR value and may be related to the duration of follow-up (Section 3.2.8).

Within studies, the increment in RR was not significantly associated with the number of dietary assessments, and was +0.07 (–0.12–0.41) (*p* = 0.52, n = 6) per 10 units GI per seven dietary assessments when estimated from NHS I, NHS II, and HPFS [10,45,60,74] by meta-regression, with the number of dietary assessments used as the covariate and with a constant for each of the three studies. This agrees with a further estimate calculated at present for these three studies based on observations in the Appendix A to Bhupathiraju et al. [10], which gave an increment in RR of 0.08 per 10 units GI per seven dietary assessment (CIs not calculable from that source). This amount corresponds to approximately a 25% increment in the RR of 1.30. Against this, the higher RR with a higher number of dietary assessments within studies is, for the present, without statistical significance and the result may not be entirely independent of increments in RR with the duration of follow-up (Section 3.2.8).

#### 3.2.10. Observations by Body Mass Index

A lack of quantitative information on the exposure to GI in studies investigating the sensitivity of the T2D–GI risk ratio to body mass index (BMI, kg/m^2^) subgroups in original prospective cohort studies prevented dose-response meta-analysis. Nevertheless, comparisons remained possible based on increments in the RR by extreme quantiles (EQM). Comparable results were found in women-only studies for the two strata of BMI. For BMI that was < 25 or <27, the combined studies’ random effects RR was 1.35 (1.01–1.80) (*p* = 0.043, n = 4) with a nonsignificant inconsistency among studies (I^2^ = 33%, τ ^2^ = 0.029, *p* = 0.21) (Appendix A). This is compared to a BMI of ≥ 25 or ≥27 when RR was 1.25 (1.12–1.41) (*p* < 0.001, n = 4) by random effects with consistent observations (I^2^ = 0, τ^2^ = 0, *p* = 0.50) (Appendix A). Insufficient evidence was available from men-only studies to assess the RR by strata of BMI, with just two studies in men: One a small study and one with an above moderate ALC intake (>15 g/d) [44,71].

Observations at higher BMI values met our criteria of a combined mean RR > 1.20 with LCL > 1.10. Observations at the lower BMI value met only the first of these two criteria.

#### 3.2.11. Studies not Eligible for the Primary Analysis

Considering the RR values for dose-response meta-analysis, the results from five publications on the T2D–GI relation were not eligible among the studies in Table 1 because they were not conducted for the longest duration of follow-up for the study. All had validity correlations for carbohydrate > 0.55. These five studies reported information from which the dose-response T2D–GI risk relations were estimated (*glst*
Section 2.2.1). The T2D–GI risk relations per 10 units GI were: 1.27 (1.08–1.49) in women-only in the NHS I study at the 6-year follow-up [74]; 1.23 (1.08–1.51) for men-only in the HPFS study at the 6-year follow-up [60]; 1.49 (1.17–1.89) for women-only in the NHS II at the 8-year follow-up [45]; 1.64 (1.48–1.83) again in women from the NHS I, but at the 24-year follow-up [10]; and 1.22 (1.00–1.44) in a European mixed-sex study at the 10-year follow-up [62] (data not tabulated). Simila et al. [75] provided insufficient information to calculate the dose-response RR and reported a T2D–GI risk ratio of 1.32 (1.01–1.72) when milk and beer were excluded as components of dietary GI.

None of the results in this section were inconsistent with the observations in Figure 3 and Table 1.

#### 3.2.12. Specific Nutrient and Non-Nutrient Factors

*(a) Nutrients*. A possibility that a dose-response T2D–GI risk relation exists due to an absence in some original prospective cohort studies of the study-level adjustment for confounding by specific nutrients was examined posteriori. For this purpose, we used the 10 valid studies (CORR > 0.55) without the outlier study [41] (Table 2). Whether GI was a surrogate for one of the specified nutrients was questioned, in which case, study-level adjustment for the specified nutrients would result in little or no association between incident T2D and GI. Study-level adjustments were made for the intakes of all fiber types (7 studies), cereal fiber (3 studies), vegetable fiber (1 study), magnesium (1 study), protein (3 studies), red meat (1 study), ALC (9 studies), energy (8 studies), saturated fat (3 studies), and trans-fats (3 studies). For all these study-level nutrient adjustments, the T2D–GI risk relation was not any lower than the risk relation given by all 10 valid studies (Table 2).

*(b) Non-nutrient factors*. Prospective cohort studies using dietary instruments with validity correlations > 0.55 for carbohydrate also yielded T2D–GI risk relations > 1.20 with lower 95% confidence limits > 1.10 when making study-level adjustments for age (10 studies), smoking (9 studies), ALC consumption (9 studies), body mass index (8 studies), physical activity (9 studies), FHD (8 studies), menopausal status and related hormone used plus oral contraceptive use (2 studies), and level of education (3 studies) (Appendix A, rows 1 to 8).

The same result was evident for the larger number of 15 studies, with both CORR ≤ 0.55 and CORR > 0.55, while adjusting for variation in both the validity correlation (centered on 0.7) and FHD (centered on 0.5) (Appendix A, rows 9 to 16).

Overall, there was no evidence that non-nutrient factors relating to incident T2D explained the size of the T2D–GI risk relation.

#### 3.2.13. Family History of Diabetes and CORR

Schulze et al. [45] observed that the T2D–GI risk relation in persons with FHD may differ from those without this history. Subsequently, a meta-analysis from Greenwood et al. [5] reported that this risk was higher in studies adjusting for a family history of diabetes. However, it is unclear whether this applies to studies with a validity correlation for carbohydrate >0.55 or whether this validity correlation remains a significant covariate when adjusting for FHD.

A meta-analysis of studies with CORR > 0.55 with clinically reported T2D and study-level adjustment for FHD gave a dose-response T2D–GI risk relation of 1.41 (1.32–1.51) (*p* < 0.001, n = 5) for which the inconsistency and heterogeneity was very low and nonsignificant (I^2^ = 5%, τ ^2^ = 0.0009, *p* = 0.379).

More studies were available when adjusting for CORR for carbohydrate as a covariate (centered on 0.7) together with FHD as a covariate (coded 1 for study-level adjustment made for FHD and 0 otherwise, centered on 0.5). The center value of 0.7 for CORR for carbohydrate was set a priori [13] as a reasonable mean value for validity correlations > 0.55 in [9]. The mean value for the present instruments with a validity correlation > 0.55 was 0.67 (SD 0.06, n = 10 studies shown in Figure 3).

The meta-analysis-level adjustment for CORR and FHD simultaneous in meta-regression showed both covariates to be significant (each with *p* < 0.001, n = 14). An adjustment for multiple testing (Monte Carlo permutations = 10,000) left CORR and FHD each significant at *p* = 0.002 and *p* = 0.001, respectively. The studies of Simila et al. [41] and van Woudenbergh et al. [72] were significant outliers (*p* = 0.009 and *p* = 0.015, respectively) while the study of Sluijs et al. [67] was inadmissible on the grounds of having no value for CORR for the study as a whole (for all their regions combined). The model explained all inconsistency and heterogeneity (I^2^ = 0%, τ^2^ = 0, n = 14 excluding the two outlier studies) and the T2D–GI risk relation was 1.28 (1.23–1.34) (*p* < 0.001, n = 14).

A higher RR arose when the covariate for FHD was centered on zero (rather than 0.5) together with the use of CORR as a covariate (remaining centered on 0.7) when RR became 1.54 (1.41–1.70) (n = 14). This again indicated a higher RR than that obtained by the primary result of 1.27 (1.15–1.40) from Table 1 and reflected the higher RR in studies applying study-level adjustments for FHD.

The significance of CORR and FHD as covariates was maintained when making an additional adjustment for the duration of follow (each *p* < 0.001, but see Section 3.2.8) or making an additional adjustment for the population average ALC consumption (Section 3.2.14).

#### 3.2.14. Alcohol

The prospective cohort study of Mekary et al. [40] showed that, within study, ALC consumption attenuates the T2D–GL risk relation, but not the T2D–GI risk relation.

Therefore, it was unexpected that additional adjustment of the model in Section 3.2.12 for the population’s average ALC consumption (centered on 7 g/d), together with the inclusion of CORR (centered on 0.7) and FHD (centered on 0.5) (Section 3.2.13) resulted in all three covariates being significant (*p* = 0.007, <0.001, and <0.001, respectively) and RR was 1.26 (1.21–1.32) (*p* < 0.001, n = 15) (I^2^ = 0% τ^2^ = 0, *p* = 0.57, n = 15) (Table 8 row 4 for GI). The study of Simila et al. [41] had a moderately high ALC consumption (11 g/d) in the range 0 to 15 g/d and was inlying due to the use of the covariate for alcohol. Meanwhile, the study of van Woudenbergh et al. [72] remained an outlier (*p* = 0.012). Assuming significant attenuation by ALC consumption, the model predicted a higher T2D–GI risk relation of 1.35 (1.27–1.43) (*p* < 0.001, n = 15) in the absence of ALC consumption and complete attenuation by 27 g/d ALC consumption when RR became 1.00 (0.85–1.19) (*p* = 0.97, n = 15) (values not tabulated).

An adjustment of the three covariate *p*-values for multiple testing (Monte Carlo permutations = 10,000) left both CORR and FHD significant (*p* ≤ 0.009), but in partial agreement with Mekary et al. [40], the attenuation by ALC was then not significant (*p* = 0.07).

#### 3.2.15. Study Size

Considering only those studies with valid dietary instruments (CORR > 0.55 for carbohydrate), study size (total number of persons followed up) in the meta-regression made no significant difference to the size of the T2D–GI risk relation (*p* = 0.29, n = 10). At the weighted mean study size (56,251 persons), the T2D–GI risk relation was 1.27 (1.15–1.39) (*p* < 0.001, n = 10) (I^2^ = 67%, τ ^2^ = 0.119, *p* = 0.001) (not tabulated). The studies analyzed were those in Figure 3 for CORR ≥ 0.55.

#### 3.2.16. Geographical Region and Ethnicity

Whether regional differences in the T2D–GI risk relation might exist was examined using Australian, Asian (east), European, and European-American (USA) study categories. For studies with validity correlations for carbohydrate > 0.55, the dose-response meta-analysis of the T2D–GI risk relation per 10 units GI was significant in all regions except Europe, which had only one study (Table 3, row 3).

The most generalizable observation is that populations with European ancestry combined met the criteria for interest for public health (RR > 1.20, RR-LCL > 1.10) for the T2D–GI risk relation (Table 3, row 5), as did persons of Eastern ancestry (Table 4, rows 6 and 7). Because the model adjusted for the validity correlation for carbohydrate, an additional East-Asian study could be included in the East-Asian ancestry category (Table 4, row 7) without the T2D–GI risk relation falling outside these criteria.

#### 3.2.17. Global Dose-Response Meta-Analysis

Global DRM was undertaken for the n = 10 studies that used dietary instruments with a validity correlation > 0.55, while withdrawing the outlier study [41] (Table 1 footnote *c* and Appendix A). Across the global range of GI from 47.6 to 76.1, the T2D–GI risk relation increased non-linearly to reach an RR of 1.87 (1.56–2.25) (*p* < 0.001, n = 10) (I^2^ = 0, τ ^2^ = 0) (Figure 5).

### 3.3. The T2D–GL Risk Relation

We searched for prospective cohort studies on the T2D–GI or T2D–GL relative risk (point estimates) or risk relations (rate estimates) that used truly valid dietary instruments (CORR > 0.55—see Methods) or could be included when CORR was used as a covariate. Among 24 reports containing relevant studies potentially meeting our inclusion criteria (Figure 1 and Figure 6), seven reports did not: One report focused on the T2D–GI risk relation only [68]. One report provided no quantitative information on exposure to GL [44]. One report did not report on the fully adjusted models [10]. One report did not conduct the longest duration of follow-up [74]. Two reports either provided no information on the validity (CORR) of their dietary instrument for carbohydrate [76] or reported this validity for only 4 of 15 regional and sex specific cohorts in a multiple regional study [67]. One study result arose from two different reports that addressed the same study, but had inconsistent dose-response results (I^2^ = 95%, n = 2) [4,40], which were precombined to represent the study once only (Section 2.1.4). The seven reports excluded are reported in further detail in the Appendix A
Section 3 (h to m). This left 17 reports, including 22 studies, available for DRM when CORR was used as a covariate. Six of the 22 studies used invalid dietary instruments (CORR ≤ 0.55) (as identified in Figure 7), leaving 16 studies available for DRM without CORR as a covariate. One study [41] with CORR considered a priori >0.55 (see [13]) was a significant outlier (*p* > 0.05) in multiple meta-analyses (Appendix A) and was withdrawn (Figure 6) only to be re-included should it fit an arising circumstance.

#### 3.3.1. Study Characteristics

The reported characteristics of the prospective cohort studies on the T2D–GL relation have been described (Appendix A). For the 15 studies using valid dietary instruments that were available to DRM and were not outliers, countries or regions included the USA (largely European-Americans,) Japan, Europe, China, Australia, and Hawaii (non-native and native separately). Sexes included men-only studies, women-only studies, and mixed-sex studies. Baseline study mean ages ranged from 38 to 75 years. All studies excluded T2D at baseline. GL in quantiles ranged from 67 to 279 g per 2000 kcal (8400 kJ) diet based on the glucose reference standard. The range of GL values across quantiles (adjusted to quintiles and to a common energy intake) in individual studies was 80 g (SD 30, minimum 39 g and maximum 146 g) per 2000 kcal (8400 kJ) diet. The number of incident T2D cases ranged from 133 to 6590 per study. Exposures were ascertained using FFQs in all studies except one, which used a diet history questionnaire [60] and one that used an undefined questionnaire plus a structured interview with a trained dietician [72]. T2D was ascertained mostly by clinical report/diagnosis or otherwise by self-report or by a mixed report (Appendix A footnote *a*). All studies adjusted for major non-nutrient factors (see Section 3.3.7 and Appendix A). The duration of studies ranged from 4 to 26 years. The average BMI values of the studies ranged from 23 to 29 or <31 kg/m^2^. Four women-only studies reported observations subgrouped by BMI, while there was an insufficient number of men-only studies to similarly distinguish between BMI subgroups.

The quality of individual studies, as assessed by the Newcastle–Ottawa score (while additionally deducting one star for CORR ≤ 0.55), ranged from 6 to 8, from possible scores of 0 to 9, suggesting that individual studies of the T2D–GL relation had little to no obvious study-level risk of bias other than for the dietary instrument. Bias to the null due to invalid dietary instruments [9,13,18,19], i.e., with a low correlation coefficient for carbohydrate versus more robust dietary records, was present in six studies when the criterion was CORR ≤ 0.55 (Figure 7). The NOS score was 7.8 (SD 0.59) for studies with valid dietary instruments and 6.2 (SD 0.98) for studies with invalid dietary instruments.

#### 3.3.2. Combined Observations

The random effects T2D–GL risk relation for studies with invalid dietary instruments (CORR ≤ 0.55) was low at 1.09 (0.97–1.23) per 80 g/d rise in GL in a 2000 kcal (8400 kJ) diet and was nonsignificant (*p* = 0.133, n = 6) (Table 4 panel 1, n = 6, random effects).

It was approximately 3 times higher at 1.26 (1.15–1.37) and significant (*p* < 0.001, n = 15) for studies with valid dietary instruments (Figure 7 and Table 4 panel 2, n = 15, random effects). This subgroup difference did not reach significance (*p* = 0.078). However, among the studies with valid dietary instruments, an association between RR and CORR was evident and significant (*p* < 0.001) (Figure 7, see also Section 3.3.5). The study of Simila et al. [41] was an outlier (*p* = 0.013, n = 15) and was therefore not included in this analysis. The study of Sluijs et al. (2010) [67] was suspected to be a high outlier (Figure 7), but failed to reach statistical significance (*p* = 0.052), hence it was retained.

#### 3.3.3. Clinical Versus Self-Report of Diabetes

Using only those studies with both valid dietary instruments (CORR > 0.55) and valid ascertainment of T2D (clinical report of T2D in ≥97% of cases) provided 10 studies of which the study of Simila et al. [41] was again an outlier (*p* < 0.042), leaving 9 studies for which the random effects RR was 1.33 (1.14–1.55) (Table 4, row 3, random effects).

#### 3.3.4. Sensitivity to Individual Studies with CORR > 0.55

After dropping studies one at a time in turn from the 15 valid studies (shown in Figure 7 with CORR > 0.55), the lowest combined random effects estimated RR was 1.21 (1.12–1.32) after dropping Sluijs et al. (2010) [62] and the highest was 1.27 (1.15–1.40) after dropping the study of Hopping et al. (2010) in Japanese-American men [42]. Hence, all such results remained >1.20 with the 95%CL > 1.10, thus meeting the criteria of interest for public health.

See also Section 3.3.8 for when multiple covariates were used.

#### 3.3.5. Dietary Instruments, a Determinant of RR

Analysis of the studies with validity correlations > 0.55 by meta-regression revealed that the T2D–GL risk relation was significantly related to CORR (*p* < 0.001) such that it reduced the inconsistency and heterogeneity to zero (I^2^ = 0, τ ^2^ = 0, *p* = 0.78, n = 15 studies).

The range of CORR > 0.55 was from 0.56 to 0.80. Across this range of CORR, the T2D–GL risk relation increased markedly by 1.54 (1.22–1.96) per 80 g GL in 2000 kcal (8400 kJ) (*p* < 0.001), indicating that CORR remained a significant and substantial source of variation in the reported T2D–GL RR values among the individual studies with validity correlations > 0.55.

When CORR was centered on 0.7 [13], the T2D–GL risk relation was 1.36 (1.26–1.48) (*p* < 0.001, n = 15) without any inconsistency (I^2^ = 0%) (Table 4 row 4). The study of Simila et al. [41] was again an outlier (*p* = 0.010). Note that the centering value of 0.7 arose a priori [13] as a reasonable value to adjust to and was based on studies published in a prior meta-analysis [9]. Among the present studies, the mean value for CORR > 0.55 was 0.67 (SD 0.07 for n = 15 studies in Figure 7).

See also Section 3.3.8 for the use of CORR in multiple covariate models and Section 3.3.16 when modelled with study-level adjustment for a family history of diabetes.

#### 3.3.6. Funnel, Trim-and-Fill, and Galbraith-Like Plot Asymmetry

A funnel plot of the T2D–GL risk relation, using studies with truly valid dietary instruments (CORR > 0.55) and without the outlier study [41], showed no evidence of asymmetry (Figure 8). The observations were adjusted to a mean value for CORR of 0.7 as deemed a priori [13] based on studies in [9]. Trim-and-fill non-parametric analysis indicated no evidence of missing studies. Thus, there was no filling of the funnel plot by trim-and-fill analysis with any hypothetical missing observations (small-study effects, such as publication bias or study selection bias). The combined non-parametric random effects dose-response T2D–GL risk relation was 1.36 (1.28–1.45) (*p* < 0.001, n = 15) (I^2^ and τ ^2^ = 0.000, *p* < 0.837), compared with the parametric value obtained by the meta-regression of log RR on CORR > 0.55 of 1.36 (1.26–1.48) (*p* < 0.001, n = 15) (I^2^ = 0 and τ ^2^ = 0, *p* < 0.78) (Table 4, row 4, random effects). Galbraith-like analysis (log RR_i_•N_i_ on N_i_, Section 2.1.7) indicated asymmetry was non-significant (Egger test, *p* = 0.97, n = 15).

#### 3.3.7. Specific Nutrient and Non-Nutrient Factors

*(a) Specific nutrients*. A posteriori, a possibility that a dose-response T2D–GL relation exists due to a failure of the original prospective cohort studies in performing study-level adjustments for specific nutrients was examined. For this purpose, as for the T2D–glycemic index risk relation (Section 3.2.12), valid studies (CORR > 0.55) were used, which, without CORR as a covariate in the random effects analysis, had a combined studies dose-response RR of 1.26 (1.15–1.37) (n = 15, *p* < 0.001) per 80 g/d GL in 2000 kcal (8400 kJ) (Table 4, row 2, random effects and Figure 7).

Among these 15 studies, for those that made a study-level adjustment for fiber (total including cereal), the T2D–GL risk relation was 1.31 (1.00–1.72) (n = 6 studies, *p* = 0.052) [40,41,45,69,71,72], cereal fiber was 1.26 (1.16–1.37) (n = 3, *p* < 0.001) [40,45,60], magnesium was 1.28 (0.78–2.09) (*p* = 0.333, n = 1) [45], protein was 2.02 (1.11–3.67) (n = 3, *p* = 0.022) [4,62,72], red meat was 1.26 (1.15–1.27) (n = 1, *p* < 0.001) [40], alcohol was 1.32 (1.14–1.53) (n = 9, *p* = <0.001), energy was 1.26 (1.14–1.39), saturated fats was 1.56 (1.02–2.37) (n = 4, *p* = 0.04), and trans-fats were 1.42 (0.94–2.14) (n = 2, *p* < 0.10) (Appendix A). No studies had a mean RR less than that for all 15 studies combined, which was 1.26 (1.15–1.37) (*p* < 0.001). The possibility that GL was a surrogate for all or any of these factors was therefore not evident.

*(b) Non-nutrient factors*. Prospective cohort studies using dietary instruments with validity correlations >0.55 for carbohydrate or carbohydrate foods yielded T2D–GL risk relations >1.20 when study-level adjustments were made for age (15 studies), smoking (9 studies), ALC consumption (9 studies), body mass index (14 studies), physical activity (13 studies), level of education (6 studies), and menopausal status and related hormone use plus oral contraceptive use (2 studies) (Appendix A, rows 1 to 8). The same results occurred with a larger number of studies (CORR ≤ 0.55 in addition to CORR > 0.55) when meta-analyses were adjusted for CORR (centered on 0.7) and FHD (centered on 0.5) (Appendix A, rows 9–16). All these results had higher RR values than the primary outcome, which was 1.26 (1.15–1.37) (Table 4, row 2, n = 15, random effects). Evidently, GL was not a surrogate for any of these factors.

#### 3.3.8. Adjustment for Multiple Covariates

A larger number of studies than the 15 studies used (Section 3.3.2, Section 3.3.3, Section 3.3.4, Section 3.3.5 and Section 3.3.6) is possible when adjusting for CORR as a covariate rather than as a group with CORR > 0.55. A meta-analysis with centered covariates to adjust for differences between the sex of participants (SEX, zero centered), two ethnicities (ETH, zero centered; European-American vs. other ethnicities), dietary instrument correlation for carbohydrate (CORR, centered on 0.7), and duration of follow-up (FUY, centered on 10 y) also reduced inconsistency (I^2^) to zero [13] (Section 3.3.5).

With these adjustments, the random effects risk relation was previously reported [13] to be 1.45 (1.31–1.61) per 100 g/d GL in a 2000 kcal (8400 kJ) diet (*p* < 0.001, n = 24), i.e., using a rounded number for g/d GL. The RR had a symmetrical funnel plot [13] and when adjusted to the lower actual dose range of 80 g/d GL, it gave an RR of 1.35 (1.24–1.46) per 80 g/d GL in a 2000 kcal diet. Two studies included in the analysis in [14] have been questioned regarding their inclusion [10] as another had used them similarly [9]. Omitting those studies [58,74] to avoid concern over possible overrepresentation of results from the NHS I made little difference to the overall result of 0.33 (1.21–1.45) per 80 g/d GL in a 2000 kcal diet (*p* < 0.001, n = 22) (I^2^ = 4%, *p* = 0.66) (Table 5, row 1).

The sensitivity of the combined studies’ RR after dropping individual studies one at a time was examined for the 22 studies available for the dose-response meta-analysis with covariates (SEX, CORR, FUY, and ETH). The highest random effects RR occurred after dropping the study of Krishnan et al. (2007) [43] after which it was 1.35 (1.24–1.47) (Appendix A). The lowest RR occurred after dropping the study of Hopping et al. (2010) (6 studies) [42], after which it was 1.29 (1.16–1.44) (Appendix A). These results are comparable to the random effects RR of 1.33 (1.21–1.45) for all 22 studies combined (Table 5 row 1). There was no evidence of the RR falling below the criteria of interest for public health (RR > 1.20 and 95% CL > 1.10).

#### 3.3.9. Sensitivity of the Fully Adjusted T2D–GL Relative Risk to Study Selections

Study selection, even with criteria at hand, remains to an extent a subjective procedure for a small proportion of available studies. For example, open to examination are: (1) The sensitivity of the RR to the inclusion of additional studies that had been excluded through a lack of information, but which might be used if uncertain assumptions are made; (2) the sensitivity to the exchange of study selections by choosing results from a different follow-up duration than the longest duration; (3) dropping studies, such as outlier studies; (4) dropping studies that might have been excluded for other reasons, such as not fully representing the population, e.g., health professionals (see Section 3.3.10). Here, we consider the sensitivity when using our fully adjusted multiple covariate model on all 22 studies, since having CORR as a covariate in all these studies became includable without significant inconsistency.

The possibilities (1 to 4 above) were examined for the dose-response T2D–GL risk relation per 80 g/d GL in a 2000 kcal (8400 kJ) diet when adjusted for covariates (SEX, ETH, CORR, FUY) (Appendix A). The results showed that the risk relation falls within a narrow range, from 1.29 (1.16–1.44) to 1.35 (1.24–1.47) compared with the 1.33 (1.21–1.45) (Table 5 row 1). All had RR values > 1.20 and 95% CL > 1.10.

#### 3.3.10. Sensitivity to Health Professionals’ Studies

The sensitivity of the T2D–GL relation to the included studies on health professionals (NHS I, NHS II, and HPS) was examined because the participants’ knowledge about health may be greater than average for the general population and this may have influenced the study performance and thus the relationship between T2D and diet. However, dropping these studies simultaneously had a negligible impact on RR (Table 6), which became 1.33 (1.20–1.48) for the 19 remaining studies instead of 1.33 (1.21–1.45) for all 22 studies. Further dropping of the outlier studies [62,64] (*p* = 0.042 and 0.023, respectively) removed the remaining inconsistency while leaving the T2D–GL relation essentially unchanged at 1.34 (1.22–1.47) (n = 17, *p* < 0.001) (Table 6).

#### 3.3.11. Observations by the Sex of Participants

After adjustment for the three hypothesized sources of inconsistencies (CORR, ETH, and FUY) (of which all three were significant), the random effects RR for women-only studies was 1.42 (1.30–1.54) (*p* < 0.001, n = 8) per 80 g/d GL in a 2000 kcal (8400 kJ) diet (Table 5, row 2 upper), which was higher than in the men-only studies at 1.23 (1.11–1.37) (*p* < 0.001, n = 6) (Table 5, row 2 lower). When considering all 22 studies together (thus including the mixed-sex studies as well), the difference between men and women was significant (*p* = 0.047) (Table 5 row 2, footnote *e*).

#### 3.3.12. Observations by Body Mass Index

Sufficient studies on the T2D–GL risk relation have been reported (Appendix A) by subgroups of BMIs (kg/m^2^) < 25 or <27 and ≥25 or ≥27 in women-only studies, though only for extreme-quintile meta-analysis because quantitative exposures (g GL and kJ energy) were not available. In the lower BMI subgroup, the combined studies random effects T2D–GL RR was 1.23 (0.99, 1.50) (*p* = 0.067, n = 4) for each 10th to 90th percentile of GL intakes. In the higher BMI subgroup, the RR was 1.33 (1.16–1.54) (*p* < 0.001, n = 4) (Table 5 row 3).

Observations of the higher BMI subgroup met the criteria of a combined mean RR > 1.20 with 95% LCL > 1.10, thus being of interest to public health. Observations at the lower BMI met only the first of these two criteria.

#### 3.3.13. Observations by Alcohol Intake

One original prospective cohort study had shown that ALC intake attenuated the T2D–GL relation in women [40]. We tested the hypothesis that differences in ALC intake might also be a source of inconsistency among the eligible studies. Intakes of alcohol often differ between men and women and this has potential to explain the sex difference in the size of the T2D–GL relation—a difference reflected again in Table 5 row 2 for women-only compared to men-only studies.

A previously used model [13] included centered covariates for (a) the fraction of participants that were women (SEX), zero centered; (b) the fraction of participants that were of European-American ethnicity vs other ethnicities (ETH), zero centered; (c) the number of follow-up years (FUY) centered on 10 years; and (d) the dietary instrument correlation for carbohydrate (CORR) centered on 0.7. When in this model participant SEX (coded as +0.5 for women and –0.5 for men) was replaced by ALC intake (ALC), which was centered on the sample population’s average of 7 g/d, where known (and coded zero when unknown). The model using ALC in place of SEX yielded a random effects RR of 1.31 (1.19–1.44) per 80 g/d GL in a 2000 kcal (8400 kJ) diet (*p* < 0.001, n = 22) (Table 5 row 5). This RR was comparable to that when SEX rather than ALC was the covariate when the random effect RR was 1.33 (1.21–1.45) (Table 5 row 1). In both models, inconsistency among studies (I^2^) was negligible at 3% to 4% (Table 5). When both sex and ALC intake were included simultaneously in the meta-analytical model, the RR was 1.31 (1.20–1.44) with I^2^ = 2% (Table 5 row 6). SEX and ALC were competitive covariates, which suggested the possibility that the sex difference in RR may be due to a sex difference in ALC consumption.

The variation in the T2D–GL risk relation with ALC consumption was clearly evident among the 11 studies where ALC intake was reported (Figure 9, *p* = 0.04). Prior to this analysis, four studies had unexpectedly low T2D–GL risk relations, three of which are included in Figure 9, with the population average ALC intake > 11 g/d (see legend to Figure 9) as they were eligible (authors had published their fully adjusted model results) and one that was not included because it was not eligible (HPS in [10]).

Possible asymmetry (small-study effects, publication bias, study selection bias) was nonsignificant (Egger test, *p* = 0.955, n = 22, Galbraith type of plot, see Methods Section 2.1.7).

#### 3.3.14. Sensitivities of RR when Using the Multi-Covariate Adjustments Including Alcohol

Sensitivities of the T2D–GL risk relation when adjusted for the population average ALC consumption, CORR, ETH, and FUY were undertaken to examine: (1) The inclusion of additional studies—for which an assumption was necessarily made about their dietary instrument correlation coefficients for carbohydrate (elevating the number of studies from 22 to 24 studies) [67,76]. (2) An exchange of results from the same study (NHS I) with a different duration of follow-up [4,40] or for studies from the same first author that may have had some common influences [62,67]. (3) The inclusion of studies with outlier observations [62,64]. (4) The exclusion of studies to which the T2D–GL risk relation was most sensitive.

Each one of the four instances (and others) had negligible influence on the T2D–GL relative risk; all remained inside the range of 1.29 (1.17–1.42) to 1.34 (1.21–1.50) (Appendix A) and so with the criteria of interest for public health.

#### 3.3.15. Protein (When a Surrogate for Carbohydrate)

One prospective cohort study hypothesized that a within-study adjustment of RR for variations in protein intake can impact on the size of the T2D–GL risk relation [10]. This was a technical issue related to the condition of simultaneous adjustments for “protein, fat, [fiber], and energy intake.” Hypothetically, such simultaneous adjustment has the potential to result in an adjustment for carbohydrate intake even in the absence of a direct attempt to do so, and might change the RR for GL towards that for GI [10], though there was no demonstration of this effect by a direct adjustment for carbohydrate intake in place of protein, and other considerations may prevail (for further detail, see endnote b after the Discussion).

An assessment of this as an additional potential influence on RR using a non-centered factor as the covariate (such as protein adjustment = 1, otherwise zero) had no or only a minor effect on RR in the model adjusted for SEX, CORR, ETH, and FUY. Thus, RR was 1.33 (1.21–1.45) per 80 g/d GL in a 2000 kcal (8400 kJ) diet before the additional adjustment (Table 5 row 1) and 1.32 (1.20–1.44) afterwards (Table 5 row 4). Thus, the primary observations made for the 22 studies (Table 5 row 1) remained robust.

However, among the present eligible 22 prospective cohort studies meta-analyzed (Table 5 row 1), only three made this conditional adjustment for protein (Table 5 row 4). Together, these three studies contributed <3% of the total weight of evidence from the 22 studies. This low contribution indicated they would have only a small influence on the combined mean RR, as found. Although the difference in the overall RR of the studies was small, the studies that made this conditional adjustment for protein had an RR greater than the combined RR of the remaining 19 studies by +0.22 (–0.15–+0.96), which is consistent with the hypothesis [10]. Support of the hypothesis was not available, though, because the increase was not significant (*p* = 0.264).

#### 3.3.16. Family History of Diabetes

Schulze et al. [45] hypothesized that the T2D–GL risk relation in persons with FHD may differ from those without this history and found the risk to be higher in those with this history in women. Subsequently, Greenwood et al. [5] observed that studies adjusting for FHD had higher T2D–GL risk relations, but without having first adjusted GL intakes for differences in the energy intake among the prospective cohort studies, and without adjustments for other covariates. Here, we adjusted the GL intakes in g/d to a common metric (80 g/d GL per 2000 kcal (8400 kJ)).

Among the studies with both valid dietary instruments (CORR > 0.55) and a valid ascertainment of exposure (clinical reports >97% of participants), the RR for studies that also made study-level adjustments for FHD was 1.61 (1.18–2.12) (*p* = 0.003, n = 4) (I^2^ = 31%, τ ^2^ = 0.0174, *p* = 0.22) (Table 7 row 4 for GL), while for those studies that did not adjust for FHD, the RR was 1.19 (1.01–1.40) (*p* = 0.039, n = 6) (I^2^ = 34%, τ ^2^ = 0.0345, *p* = 0.21) (not tabulated). This difference did not reach statistical significance (*p* = 0.11, n = 10 studies).

More studies were available when adjusting for the validity correlation. Seven studies from undertook study level adjustment for FHD and were [45,46,60,62,69,72] and [4] precombined with [40]. When centering the covariate for CORR on 0.7, but the covariate for FHD on zero (7 studies coded 0 and the remainder coded 1), a result for studies undertaking study-level adjustment for FHD was obtained for which the RR was 1.40 (1.25–1.56) (*p* < 0.001, n = 21) (I^2^ = 0%, τ^2^ = 0.000, *p* = 0.513), indicating that studies undertaking study-level adjustment for FHD gave a higher RR value than those that did not, for which the RR was 1.29 (1.19–1.40). The difference made by the covariate for FHD was not significant (*p* = 0.149).

CORR alone as a covariate was significant for the eligible 22 studies (*p* = 0.006), the 21 studies (*p* < 0.001, Simila et al. outlier (*p* = 0.016)), and 15 studies (*p* < 0.001) without the studies using a dietary instrument with a validity correlation ≤ 0.55, and without the outlier study from Simila (*p* = 0.010). CORR remained a significant covariate when, in addition, a covariate for study-level adjustment for FHD was applied; for the n = 22, n = 21, and n = 15 conditions, the probabilities for the covariate CORR were *p* = 0.002, *p* < 0.001, and *p* = 0.001, respectively. Meanwhile, study-level adjustment for FHD was never significant (*p* = 0.133, *p* = 0.149, and *p* = 0.851, respectively).

Considering now the T2D–GL risk relation with the 21 studies and after adjustment for CORR (centered on 0.7), the random effects RR was 1.32 (1.22–1.43) (*p* < 0.001, n = 21) (I^2^ = 1%, τ ^2^ = 0.0002, *p* = 0.64) (Table 8 row 1 for GL). Further adjustment for FHD (centered on 0.5) made little difference as RR was 1.34 (1.24–1.46) (*p* < 0.001, n = 21) (I^2^ = 0, τ ^2^ = 0, *p* = 0.64) (Table 4 row 5 and Table 8 row 3 for GL). Further adjustment for the average ALC intake (centered on 7 g/d) also made little difference as RR was 1.35 (1.22–1.49) (Table 8 row 4 for GL)

Adjustments for CORR alone could reduce inconsistency and heterogeneity to near zero (Table 8 row 2 for GL, n = 15 studies); this was also true when including studies with a low validity correlation for carbohydrate (Table 8 row 1 for GL, n = 21 studies). Compared with the covariate of CORR, FHD was not dominant in affecting the size of the T2D–GL risk relation. By contrast, when adjusting the T2D–GI risk relation for CORR, FHD was a significant factor (Section 3.2.13).

#### 3.3.17. Studies not Eligible for the Primary Analyses

Considering the T2D–GL risk relations for dose-response meta-analysis, the results from four publications were not eligible among the studies in Table 4, Table 5, Table 6, Table 7 and Table 8 because they did not use the longest duration of follow-up or had missing information of which assumptions had to be made.

Salmeron et al. (1997) in women [65] (NHS I at the 6-year follow-up) had a dose-response T2D–GL risk-relation estimate of 1.65 (1.18–2.29) per 80 g/d GL in a 2000 kcal (8400 kJ) diet without adjustments for the covariates of SEX, CORR, ETH, and FUY, and 1.49 (1.07–2.07) after making adjustments for SEX (equal numbers of males and females), CORR (centered on 0.7), ETH (centered on equal numbers of European-American and non-European--American ethnicities), and FUY (centered on a 10-year follow-up).

Sluijs et al. (2013) [67] reported on CORR for only 4 of the 15 included cohorts in their multi-regional study. The dose-response T2D–GL risk relation was estimated at 1.14 (0.87–1.49) per 80 g/d GL in a 2000 kcal (8400 kJ) diet without adjustment for SEX, CORR, ETH, and FUY. After adjustment for these coveriates, and assuming a value for CORR of 0.5 for this study, an adjusted RR of 1.40 (1.07–1.83) was identified.

Rossi et al. [76] estimated a dose-response T2D–GL risk relation of 1.25 (0.92–1.67) without adjustment for SEX, CORR, ETH, and FUY. After adjustment, this was 1.58 (1.17–2.14) assuming CORR was 0.45, which was the average of the separate values for polysaccharides and sugar, in men and women, rather than specifically for carbohydrate in the corresponding validation study [77].

The study of Bhupathiraju et al. [10] reported three studies for which the results of the fully adjusted model were not reported study-by-study, but as the three studies combined by fixed effects meta-analysis. This allowed an estimation of the dose-response T2D–GL risk relation for their three studies combined, which was 1.39 (1.18–1.63) per 80 g/d GL in a 2000 kcal (8400 kJ) diet without adjustment for SEX, CORR, ETH, and FUY, and 1.31 (1.12–1.53) after adjustment.

Simila et al. [75] published an extreme quantile T2D–GL risk ratio of 1.32 (0.85–2.07) for the T2D–GL risk relation when including beer and milk as a component of the dietary GL (insufficient information was available to estimate the risk relation per 80 g/d GL in a 2000 kcal diet).

Consistent with the low sensitivity of the combined studies’ T2D–GL risk relation to study selection (Section 3.3.9 and Section 3.3.10), none of these studies disagreed with the T2D–GL risk relations reported in Table 5, Table 6, Table 7 and Table 8.

#### 3.3.18. Geographical Region or Ethnicity

Studies on persons of European ancestry (the USA, EU, Australia) combined for CORR > 0.55 yielded a T2D–GL risk relation of 1.35 (1.17–1.55) (*p* < 0.001, n = 10) (I^2^ = 30%, τ^2^ = 0.013, *p* = 0.166) (not tabulated), including studies from Salmeron et al. in men [60], Schulze et al. [45], Sahyoun et al. [70], Patel et al. [61], Hopping et al. (2 studies) (FCA, and MCA) [42], Sluijs et al. [62], van Woudenbergh et al. [72], and the precombined study form Mekary et al. and Halton et al. [4,40]

Studies from persons of Eastern ancestry (China, Japan, native Hawaii) for CORR > 0.55 yielded a T2D–GL risk relation of 1.19 (1.05–1.34) (*p* = 0.007, n = 5) (I^2^ = 44%, τ^2^ = 0.001, *p* = 0.008) (not tabulated), including studies from Villegas et al. [46], Sakurai et al. [71], and Hopping et al. with three studies (MJA, MNH, FNH) [42]. However, after adjustment for the validity of the dietary instrument (size of CORR centered on 0.7), the T2D–GL risk relation for Eastern ancestry was 1.32 (1.18–1.49) (*p* < 0.001, n = 5) (I^2^ = 0, τ^2^ = 0.0000, *p* = 0.95) (not tabulated). In this model, the covariate CORR was significant (*p* = 0.009). Thus, the apparent lack of meeting the criteria of RR > 1.20 and a lower limit of > 1.10 in the persons of Eastern ancestry was due to the use of dietary instruments with lower validity, CORR > 0.55, but <0.7, except for the study from China [42], which applied a dietary instrument with CORR = 0.71 and, without adjustment for CORR, had a T2D–GL risk relation of 1.36 (1.17–1.58) (n = 1 study, 64,227 persons, *p* < 0.001).

Persons of European ancestry had a T2D–GL risk relation > 1.20 with a lower 95% CL > 1.10 without having to adjust for covariates. Except for persons from China who met these criteria, persons of Eastern ancestry met this criterion only after adjustment for the validity correlation centered on 0.7.

#### 3.3.19. Global Dose-Response Meta-Analysis

Global DRM was undertaken for studies that used valid dietary instruments for carbohydrate (CORR > 0.55) while withdrawing the outlier study [41] in meta-analyses from multiple perspectives (Appendix A), leaving n = 15 studies. Across the global range of GL from 73 to 257 g per 2000 kcal (8400 kJ), the T2D–GL risk relation increased non-log linearly to reach an RR of 1.89 (1.66–2.16) (*p* < 0.001, n = 15) (I^2^ = 0, τ ^2^ = 0) (Figure 10), with the confidence limits based on the three error sources (τ^2^ + σ^2^ + σ_f_^2^).

## 4. Discussion

1. Our quantitative dose-response meta-analyses indicate strong associations between markers of the postprandial glycemic effect of foods (both GI and GL) and the development of T2D relative to criterial interest in public health nutrition. We analyzed prospective cohort studies that used population specific truly valid dietary instruments for the estimation of exposure. The risk relations were >1.20 with lower 95% confidence limits > 1.10 for both GI (per 10 units GI) and GL (per 80 g/d GL in a 2000 kcal (8400 kJ) diet) (Table 1 and Table 4). We also conducted meta-analyses of studies using dietary instruments that were considered invalid by the criterion used. Consistent with the invalidity, the RR values were not different from zero (Table 1 and Table 4). These intake ranges corresponded to approximately the 10th and to the 90th percentiles of the population intakes averaged across the sampled populations within jurisdictions as assessed using the truly valid dietary instruments.

2. Among the studies using valid dietary instruments, the T2D–GI risk relation was 1.27 (1.15–1.40) (*p* < 0.001, n = 10 studies), which was our primary outcome for GI (Table 7 row 2 for GI). Potentially, this was an underestimate because a subgroup of studies that used the most valid method of ascertainment of T2D (clinical report) showed a higher risk relation of 1.40 (1.29–1.51) (Table 7 row 3 for GI) and the risk relation remained higher for a subgroup that both used clinical report and made study-level adjustments for FDH when RR was 1.41 (1.32–1.51) (Table 7, row 4 for GI).

3. The outcome for the T2D–GL risk relation was 1.26 (1.15–1.37) (*p* < 0.001, n = 15 studies), which was our primary outcome for GL (Table 7 row 2 for GL). Again (cf. above paragraph for GI), this was potentially an underestimate because a subgroup of studies ascertaining T2D by clinical rather than self-report yielded a higher RR of 1.33 (1.14–1.55) (Table 7 row 3 for GL) and the risk relation remained higher in a subgroup that both used clinical report of T2D and made study-level adjustments for FHD, with an RR of 1.61 (1.18–2.12) (Table 7 row 4 for GL).

4. Other than for the statistically significant (*p* < 0.05) outlier studies, essentially, all inconsistency among studies was explained by the covariates for CORR and FHD, when inconsistency was not evident (I^2^ = 0%) for both the GI and GL risk relations (Table 8, row 3 for GI, and row 3 for GL).

5. Risk of bias at the outcome level was minimal. Thus, the trim-and-fill meta-analyses agreed with the primary results and identified a zero number of studies required to achieve symmetry in funnel plots. Galbraith like-plots (log RR•N on N) similarly indicated no significant bias according to the Egger test. These two plots for GI and GL indicated that there was no evidence of missing studies, such as publication bias or study-selection bias. Review-level systematic bias was more likely due to an underestimation of the risk relations rather than overestimation since studies of higher specification in respect of the use of clinical reports for the ascertainment of T2D and adjusted for a family history of diabetes, even when selecting for studies using dietary instruments with a validity correlation for carbohydrate >0.55 (as noted above), yielded high RR values.

6. The adjustment to RR using the validity correlation for carbohydrate (CORR) as a covariate enabled those studies using inferior dietary instruments (CORR ≤ 0.55) to be included in secondary outcomes to obtain an adjusted combined studies RR value for a predefined value for CORR and to explain the remaining variation in RR related to CORR. This was limited by the need for prior identification of a correlation value to center upon. A prior mean value of 0.71 (SD 0.06, n = 4 studies) was calculated from the validity correlation > 0.55 from the very first meta-analysis on the T2D–GI and T2D–GL risk relations, together with a value of 0.69 (SD 0.07, n = 15 studies) for other disease GI and GL risk relations in [9]. Prior values of 0.71 (SD 0.05, n = 6 studies) and 0.73 (SD 0.05, n = 6 studies) were obtained for studies on men and women, respectively, for studies with CORR > 0.55 when using combined studies of coronary heart disease-GI and GL risk relations [19]. In the present paper, for the primary outcomes, the values obtained were 0.67 for GI (SD 0.06, n = 10 studies in Figure 3) and 0.67 for GL (SD 0.07, n = 15 studies in Figure 7). A value of 0.70 to center upon therefore appears to be a reasonable value and was used previously for the T2D–GL risk relation [13]. When centering CORR on 0.70, a T2D–GL risk relation adjusted for covariates (CORR, SEX, ETH, and FUY as in [14]) yielded a risk relation of 1.33 (1.21–1.45) per 80 g/d GL in a 2000 kcal diet among n = 22 studies (Table 5, row 1), which is a revision on that reported previously [14].

7. For valid studies (using dietary instruments with validity correlations for carbohydrate >0.55), there was no difference in the T2D–GI risk relation between men-only and women-only studies (Table 1). Likewise, there was no difference in the size of the T2D–GL risk relation between women-only and men-only studies (when adjusting for CORR and FHD) (Table 4 rows 6 and 7). By contrast, there was a significantly higher T2D–GL risk relation in women than in men after adjustment for the dietary instruments’ correlation coefficients for carbohydrate, ethnicity, and duration of follow-up (*p* < 0.030, n = 14) (Table 5, row 2, and footnote *e*). Nevertheless, in summary, both men and women had T2D–GI and T2D–GL risk relations that were >1.20 with lower 95% confidence limits > 1.10.

8. Evidence from this and prior meta-analyses [5] suggests, from the studies analyzed, that the T2D–GI risk relation may be significantly higher with a longer duration of follow-up and with increased applications of a dietary instrument during follow-up [10]. However, this difference was neither significantly different when analyzed within studies (Section 3.2.8, Section 3.2.9) nor significantly different for the T2D–GL risk relation with an increased duration of follow-up in the multi-covariate model [13]. However, increased applications of a dietary instrument applied within studies will in principle improve the capturing of a representative diet over time and will account for changes in the diet that may occur during the long durations of a study, which, if deviating from basal intakes, would be expected to be biased toward a null RR [22].

9. Both the T2D–GI and T2D–GL risk relations were attenuated at higher than lower average population ALC consumption (Section 3.2.14 and Section 3.3.14). This was evident also for incident coronary heart disease among prospective cohort studies in a recent publication [19]. A within-study result in women also identified an attenuation of the T2D–GL risk relation with higher ALC consumption, but not significantly or at all for the T2D–GI risk relation [40]. An implication is that our primary estimates of the T2D–GI and T2D–GL risk relations may underestimate the risk relation for persons not consuming ALC. This needs further investigation within studies.

10. Our global dose-response meta-analyses provide information of importance regarding the worldwide risk of incident type 2 diabetes. The RR was 1.87 (1.56–2.25) (*p* < 0.001, n = 10) for the T2D–GI risk relation from 47.6 to 76.1 units GI (Figure 5, Section 3.2.17) and 1.89 (1.66–2.16) (*p* < 0.001, n = 15) for the T2D–GL risk relation from 73 to 257 g/d GL in a 2000 kcal (8400 kJ) diet (Figure 10, Section 3.3.19). These relations were not log-linear, implying a curvature in the global dose-response plots, which may be explained by the heterogeneity among studies. Consistent with this, the local dose response plots for GI and GL fitted log-linear relations as shown in our companion paper [15]. The global observations imply that persons consuming a diet with a GI of 76 could have an 87% higher risk of developing diabetes than persons consuming a diet with a dietary GI of 48. Likewise, persons consuming a diet with a GL of 257 g/d in a 2000 kcal could have an 89% higher risk of developing diabetes than persons consuming a diet with a GL of 73 g/d in 2000 kcal.

11. The above can be put further into context regarding the range of GI values within heathy food groups (wholegrain, fruit, vegetables, etc.). For each of these food groups, the GI spans across approximately 60 units of GI [78], which is nearly twice that of 28 units of GI across diets, from the lowest (48 units GI) to the highest exposure (76 units GI) during global risk assessment (Figure 5). Thus, extremes of food GI values within healthy food groups might pose a nearly 4-fold difference in risk for T2D (i.e., 3.8 = exp[log(1.87) × 60/28]). Expressed another way, this is nearly a 400% difference in risk between the lower and upper extremes of the GI within ‘healthy’ food groups. This is a substantial percentage to be wary of when choosing foods, particularly that there is a similar risk to be wary of with regard to CHD [19]. If one is not willing to be wary of high GI foods, then one can, in this context, confidently select lower GI foods from within food groups given the confidence in causation [15].

12. A priori dose-response meta-analyses of the T2D–GI and T2D–GL risk relations [5] indicated the combined studies T2D–GI and GL risk relations were dissimilar, with GI being the stronger risk factor. However, when using only studies with truly valid dietary instruments and after adjustment of the GL to a common metric (2000 kcal diet), we found that the two risk relations (for GI and GL) were similar when analyzed by local or as global DRM, as noted above. In comparing the relative strengths of the T2D–GI and GL relations, it should be recognized, however, that it is not a question of which is better. GI and GL may find preference according to the situation of the person or patient [79,80].

13. A strength of our analyses is that we undertook dose-response meta-analysis [20] to provide quantitative information, allowing an estimation of the risk relations from the 10th to the 90th percentile of exposure averaged across the sampled populations. This allowed a comparison with the threshold relative risks recommended for triggering concern for the long-term health of the general (healthy) population [15,17]. The quantitative dose-response meta-analysis also allowed a synthesis of the global dose responses, which gave a wider perspective on the importance of GI and GL than can be reached by the dose-response across typical intakes within regions or countries (jurisdictions) alone or by extreme quantile meta-analysis, and also uses more of the available RR data from studies than EQM does.

14. A further strength is that our primary outcomes control for the validity of the dietary instruments used in the original studies, for which there is a need to limit the study-level performance bias [9,13,18,19]. Such control agrees with the view [9,13,19,24] that care is needed when interpreting the meta-analyses of observational studies, especially to not simply combine the results of all known studies. The importance of high quality dietary instruments is very clear in nutritional epidemiology [22], but unfortunately, while a standard has been suggested [18], it is also clear that authors often ignore this, instead claiming validity of a dietary instrument seemingly just because the instrument has been put through a validation procedure rather than having met some predefined criterion for validity. A consequence of this for the T2D–GI and GL risk relations is that some studies have reported highly attenuated risk relations (Table 1 and Table 4, rows 1 vs. rows 2); this was evident also for incident CHD in relation to GI and GL [19] and other diseases [9].

15. In more detail, there is no step change in the attenuation of these risk relations at CORR = 0.55. Instead of deselecting studies with CORR < 0.55, it was possible in some instances to adjust for variation in CORR among studies by using it as a covariate centered on 0.7, which was the approximate mean for CORR values > 0.55. This is reasonable when CORR is a significant covariant or becomes so when other factors are considered. The T2D–GL risk relation was associated with CORR across all ranges (≤0.55 and >0.55) (*p* < 0.001, Section 3.3.5 and [13]), whereas the T2D–GI risk relation was significantly associated with CORR (*p* < 0.001) only when adjusted simultaneously for FHD subgroups, which was also a significant factor (*p* < 0.001) (Section 3.2.13). The application of adjustments for both CORR and FHD reduced the inconsistency to ≤1% for both the T2D-GI and T2D–GL risk relations (Table 8, row 3 for GI and row 3 for GL). In these, the validity correlation was the dominant factor, especially for GL.

16. A further strength was that studies specified to be excluded, because they did not use the longest duration of follow-up, also had dose-response T2D–GI and GL risk relations analyzed as >1.20 (Section 3.2.11. and Section 3.3.17). Thus, the selection process, for eligible observations of the longest follow duration within a study, did not inadvertently compromise our findings.

17. Further, we asked whether high T2D–GI and GL risk relations might arise from a failure to adjust for other dietary factors, e.g., fiber, magnesium, protein, and red meat, which may be associated with the incident of T2D and for which GI and GL may have been surrogates. However, sufficiently high T2D–GI and GL risk relations (>1.20) was found in studies that utilized study-level adjustment for each of these factors (Section 3.2.12 and Section 3.3.7), which adds strength to our findings.

18. Among other strengths, our analyses used only prospective cohort studies, which are the most reliable type of observational study when a dietary instrument is truly valid.

19. A weakness is that the validity of a dietary instrument had to be based on the validity correlation coefficient for carbohydrate or carbohydrate foods rather than more directly on the validity correlations for GI and GL. This was because almost all studies and accompanying dietary instrument publications did not report validity correlations for GI or GL. Notably though, data from Barclay et al. [68,81] support some similarity among the validity correlations for GI and carbohydrates.

20. Another weakness is that the application of GI and GL values to foods has been of concern. In epidemiology, there is now no concern about the analytical accuracy of food GI and GL values [82]. More concern exists about the accuracy of the assignment of GI values to local foods eaten when attempting to match them to foods in the international tables of GI and GL [83]. This weakness could lead to the misclassification of foods to GI and GL categories (quantiles), which traditionally has been thought to cause bias towards a null relative risk (null for RR = 1) [84,85], but which may add to the heterogeneity among study estimates of risk relations due to both over- and underestimation of the relative risks among studies [86]. Under conditions of low or nil random error (Table 8), it is more likely that the risk relations are underestimated than overestimated due to such misclassification [86]. Further, because we undertook dose-response meta-analysis, any misclassification broadening the range of GI and GL intakes would further increase the likelihood of bias toward the null hypothesis for the reasons noted in the following paragraph.

21. A bias that is not well recognized in prior meta-analyses of the dose-response T2D–GI and GL risk relations [5,13] (or any other nutrients for that matter) is a particular bias that arises when using quantitative dose-response meta-analysis whenever an estimation of food consumption is imperfect. Dietary instruments that do not collect representative estimates of food consumption yield overdispersed estimates for the distributions of nutrient intakes [22], which can lead to marked underestimation of the relative risk by the means already noted in the introduction.

22. The high level of inconsistency (I^2^) among studies for the T2D–GI risk relation without adjustment for related factors (Table 1) may seem to be of concern, but was attributable to variation in CORR among studies and whether adjustments had been made at the study-level for FHD (Table 8 row 3 for GI) or used clinical reports for the ascertainment of diabetes (Table 7 row 3 for GI). The nonlog linearity in the global dose response curves associating the incident of T2D with GI (Figure 5) could also be explained by variation in the validity correlation and subgroups for FHD. Importantly though, the threshold criteria for interest to public health nutrition (RR > 1.20 with a >1.10 lower 95% confidence limit for harmful exposure when the referent exposure is the lowest exposure category) make no requirements about the level of inconsistency or heterogeneity, merely that this does not lead to the lower 95% limit criterion (>1.10 in this instance) being unsatisfied.

23. Concerning public health relevance at the global level, our evidence indicates that GI and GL are substantial food markers for the prediction of the development of T2D worldwide (Section 3.2.17. and Section 3.3.19). Persons of European ancestry had significant risk relations (*p* < 0.001 for both GI and GL), which met both criteria of interest for public health (RR > 1.20 and lower 95% CL > 1.10). Persons of Eastern ancestry (China, Japan, and native Hawaii) were also at significant risk, but met neither criterion, except the one study from China or unless the risk relations were adjusted for CORR as a covariate (Section 3.2.16. and Section 3.3.18). The need for this adjustment suggests a performance bias towards the null due to the use of dietary instruments with low validity correlations for carbohydrates. With this adjustment, persons of both ancestries met the two criteria of interest for public health. The limited evidence for high T2D–GI and GL risk relations from within Europe itself contrasts with the wealth of evidence from European studies showing high CHD–GI and GL risk relations, particularly for GL [19]. In part, this was because of the application of dietary instruments with slightly higher validity correlations when investigating the CHD than the T2D–GI and GL risk relations.

24. Looking to future research needs or gaps in research, it is uncertain whether people outside the above regions would avoid these risks, e.g., Africa, South America, possibly India and others, and their diaspora. Further, research is also needed to establish whether nonconsumers of ALC are at a greater risk than ALC-consumers, particularly among men; this might begin with asking whether dietary instruments have the same level for validity across different levels of ALC consumption [87]. With regard to body mass index (BMI, kg/m^2^), women of the higher BMI strata (>25 or 27) met the criteria of concern for public health nutrition (RR > 1.20, L95% CL > 1.10) for both GI and GL (Section 3.2.10 and Section 3.3.12), but information for men was insufficient to draw a conclusion. Thus, more information on susceptible subgroups, including by body mass index and by family history of diabetes, would be of interest at the subgroup level, again particularly in men.

## 5. Conclusions

Critical analyses of prospective cohort studies provide robust evidence that diets higher in glycemic index (GI) and load (GL), independently of dietary fiber, substantially elevate the risk of type-2 diabetes among healthy populations of men and women, each sufficiently meeting the criteria of importance to public health nutrition. This is supported by the global (worldwide) incidence of diabetes increasing by approximately 90% across the global range of exposures, and in a companion paper [13], by the Bradford–Hill criteria indicating that these relations are probably causal.

## 6. Endnotes

**^a.^** In a recent consensus summit [88], the glycemic response (GR), glycemic index (GI), and glycemic load (GL) were defined as follows: “GR is the post-prandial blood glucose response (change in concentration) elicited when a food or meal that contains carbohydrate is ingested. [...] The GI is conceptually the GR elicited by a portion of food containing 50 g (or in some cases 25 g) of available carbohydrates and is expressed as a percentage of the GR elicited by 50 g (or 25 g) of the reference carbohydrate. [...] The GI is therefore both a standardized GR (based on an equal amount of available carbohydrates) and a relative GR (relative to a referent food). It is a property of the food itself, an index or percentage representing a quality of carbohydrate foods. […] The GL is the product of the GI and the total available carbohydrate content in a given amount of food (GL = GI x available carbohydrate/given amount of food). […]. Thus, depending on the context in which GL is used, the GL has … units of g per serving, g per 100 g food, and g per 1000 kJ or 1000 kcal.” Because GL is a product of GI, it too ranks foods by their glycemic response rather than predicting the actual response within or across persons. Note that adding protein or fat or fiber to a carbohydrate food, to create another food, can sometimes modify the glycemic response as noted in reviews [88,89,90,91], which indicates that the GI value pertains to the food item or meal rather than solely to the carbohydrate it contains.

**^b.^** When, in original prospective cohort studies, study-level adjustments are made simultaneously for variations in energy intakes and all individual macronutrient intakes other than carbohydrate, then under ideal circumstances, adjustments are automatically also made for variations in carbohydrate intake by difference. Thus, when examining a disease–GL risk relation and holding carbohydrate intake constant, the variation in GL is considered due to variation in GI [10]. There are, however, several potential confounders of this possible interpretation: Firstly, protein may have effects independently of its energy or bulk content. Second, a strong interaction has been suggested between the energy:protein ratio and glycemic index (Livesey, G. 2015, 33rd Diabetes and Nutrition Study Group meeting, Toronto), which is not considered. Interaction or additivity between protein foods and carbohydrate quality is also suspected [92,93]. Lastly, third, cumulative error may arise from combining errors in the dietary instrument for each of the protein, fat, and fiber (and alcohol) intake; thus, the cumulative error may have unpredictable effects.

## Figures and Tables

**Figure 1 nutrients-11-01280-f001:**
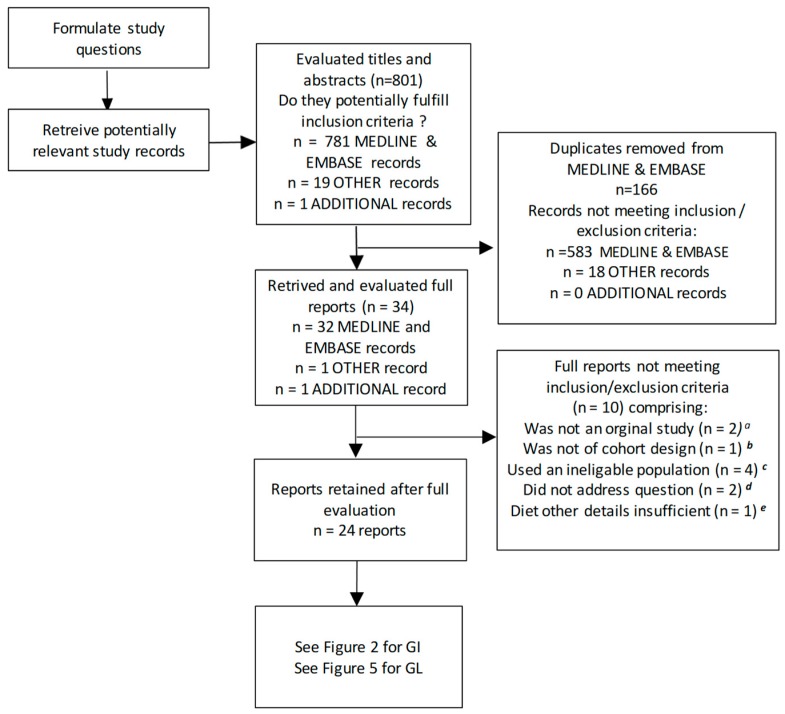
Process of the review and outcomes of the inclusion/exclusion criteria for the T2D–GI and GL risk relations. The literature was searched for prospective cohort studies investigating these relations, from 1946 to 6 December 2018. MEDLINE, EMBASE, and other sources (see Methods) were searched. ***^a^***–***^e^***: See Section 3 of the Appendix A for further details. Abbreviations: GI, glycemic index; n, number of published reports; T2D, type 2 diabetes; GL, glycemic load.

**Figure 2 nutrients-11-01280-f002:**
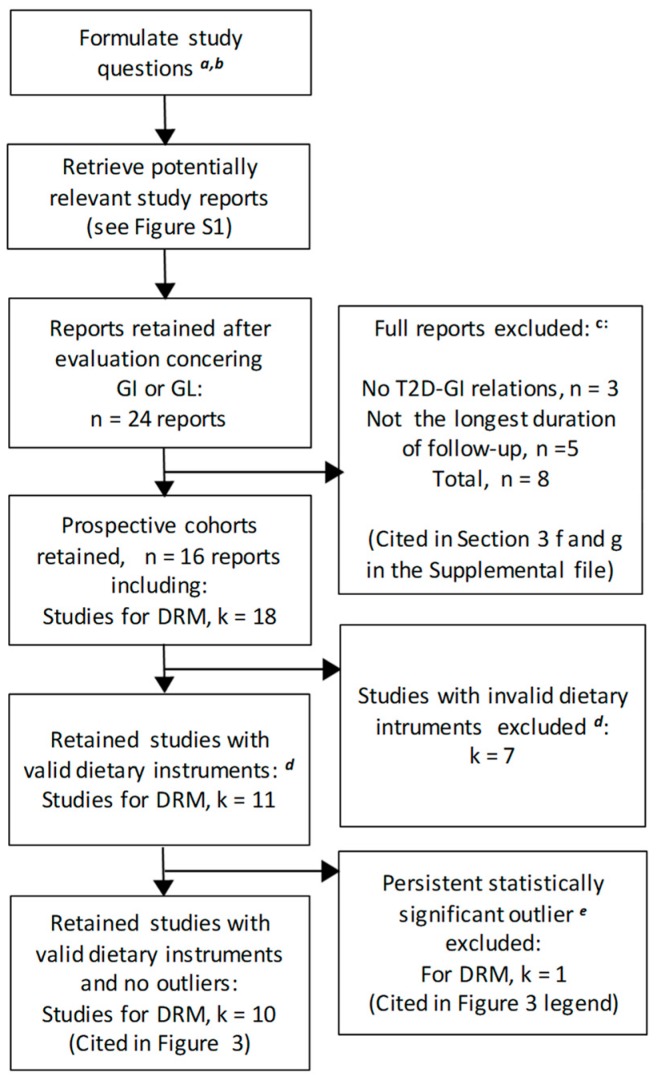
Process of the review and outcomes of the inclusion/exclusion criteria for the T2D–GI risk relation. The literature was searched for prospective cohort studies investigating this relation, from 1946 to 6 December 2018. MEDLINE, EMBASE, and other sources (see Methods) were searched. Superscripts: ***a***. Whether T2D is related to the dietary glycemic index (GI) and whether the risk relation is >1.20 with a lower confidence limit > 1.10 for each 10th to the 90th percentile of GI intake (which on analysis was per 10 units GI). ***b***. Is the T2D–GI risk relation conditional on the validity of the dietary instrument, on the sex of the participants, or on the BMI (kg/m^2^)? ***c***. Details of the full reports excluded are described in the associated text and in more detail in Section 3 (***f*** and ***g***) of the Appendix A. ***d***. Studies with dietary instrument correlations for carbohydrate ≤ 0.55 were invalid. ***e***. When a study was suspected of being an outlier was proven to be so upon analysis (*p* < 0.05) and persisted to be so in several different meta-analyses (Appendix A), it was excluded from the primary analysis. Abbreviations: DRM, dose-response meta-analysis; GI, glycemic index; n, number of published reports; k, number of studies (more than one may appear in one report); T2D, type 2 diabetes.

**Figure 3 nutrients-11-01280-f003:**
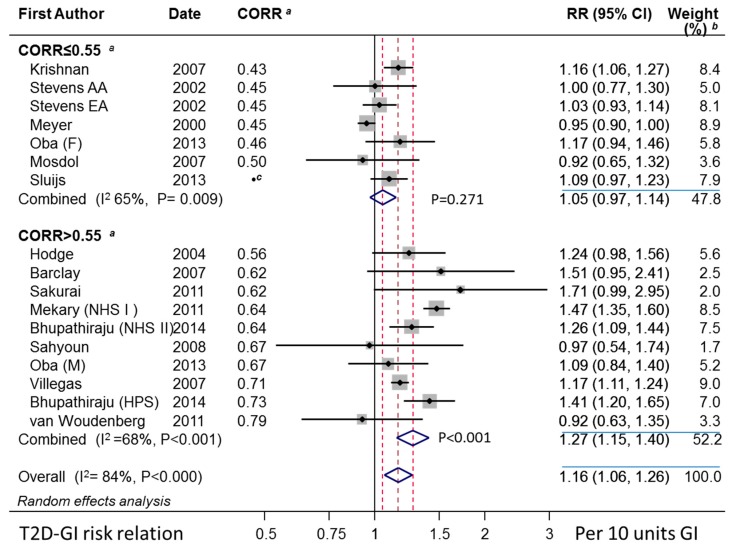
Forest plot of the dose-response T2D–GI risk relation by validity of the dietary instrument. The box size is proportional to the weight contributed by the study to the combined study mean. Horizontal lines span individual study 95% confidence intervals. Arrowheads indicate truncations. Diamonds represent the combined study mean RR values per 10 GI units (based on the glucose standard), and the corresponding 95% CI values. Analysis was undertaken on the natural logarithms and are shown untransformed. *p*-Values were calculated using the z test for RR and the Chi^2^ test for I^2^. Eligible studies were: Barclay et al. (2007) [68], Bhupathiraju et al. (2014) (NHS II) [10], Bhupathiraju et al. (2014) (HPS) [10], Hodge et al. (2004) [69], Krishnan et al. (2007) [43], Mekary et al. (2011) [40], Meyer et al. (2000) [64] (*p* = 0.011), Mosdol et al. (2007) [65], Oba et al. (2013) (men) [44], Oba et al. (2013) (women) [44], Sahyoun (2008) [70], Sakurai et al. (2012) [71], Sluijs et al. (2013) [67], Stevens et al. (2002) (AA) [66], Stevens et al. (2002) (EA) [66], van Woudenbergh et al. (2011) [72], and Villegas et al. (2007) [46]. An outlier study was: Simila et al. (2011) [41] (*p* = 0.033). Abbreviations: AA, African-American; EA, European-American; CI, confidence interval; HPS, Health Professionals Study; I^2^; inconsistency; NHS, Nurses’ Health Study; *p*, probability; RR, risk relation. Footnotes: *a*, CORR refers to the dietary instrument’s correlation coefficient for carbohydrate or carbohydrate foods; *b*, percentage weights are from random effects; c, the study of Sluijs et al. [67] reported no overall correlation for their combined regions, some regions had correlations <0.55.

**Figure 4 nutrients-11-01280-f004:**
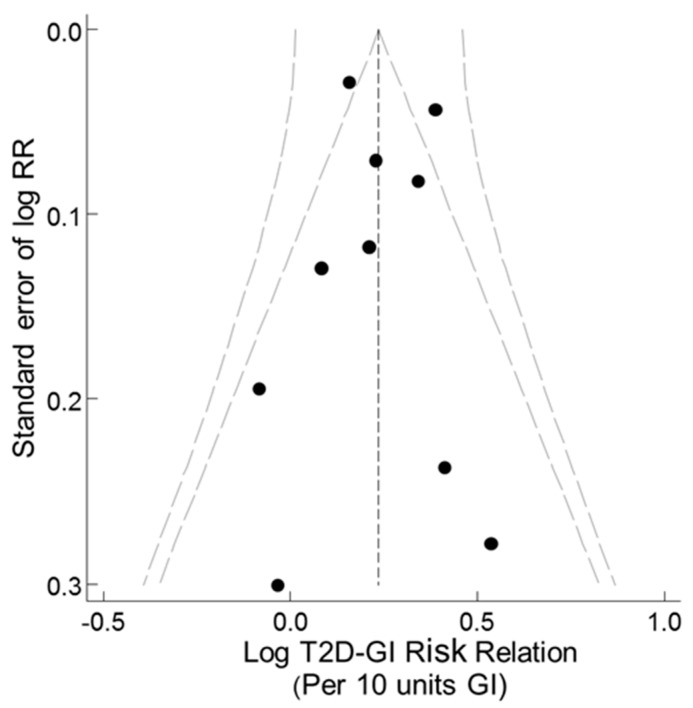
Funnel plot of the study results for the T2D–GI risk relation for valid studies using valid dietary instruments. Valid studies were those with valid dietary instruments (correlation coefficient for carbohydrate or carbohydrate foods >0.55, deemed a prior) and not including an outlier study [38]) (Appendix A) (*p* < 0.05, deemed a priori in the random effects model. The inverted funnel shows the study-level 95% confidence limits with fixed effects, the outer funnel shows the study-level 95% confidence limits with random effects, and the vertical dashed line shows the combined studies random effects trim-and-fill mean, which exponentiated was 1.27 (1.15–1.40) (*p* < 0.001, n = 10 studies).

**Figure 5 nutrients-11-01280-f005:**
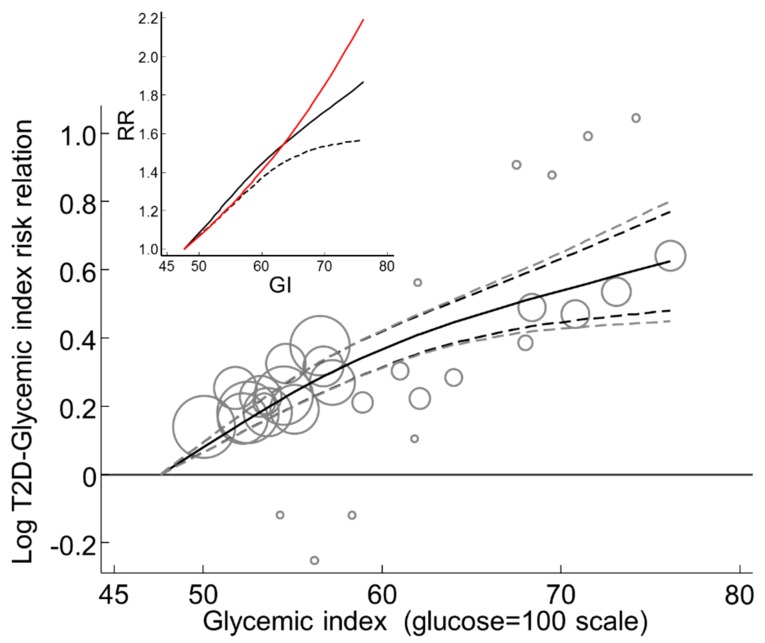
Global dose-response meta-analysis of the T2D–GI risk relation. The larger plot shows the mean increment in the log risk relation and 95% CI values. The 95% CI values were based on random effects analysis, the inner (black dashed) lines show the conventional CI values based on (τ^2^ + σ^2^) whereas the outer (grey dashed) lines were based on an additional forecasting term, σ_f_^2^, so that the outer 95% CI was based on (τ^2^ + σ^2^ + σ_f_^2^). Insert: Shows the corresponding unlogged RR (black, continuous), the lower 95% confidence limit when accounting for σ_f_^2^ (black dashed), and the corresponding log-linear prediction (red).

**Figure 6 nutrients-11-01280-f006:**
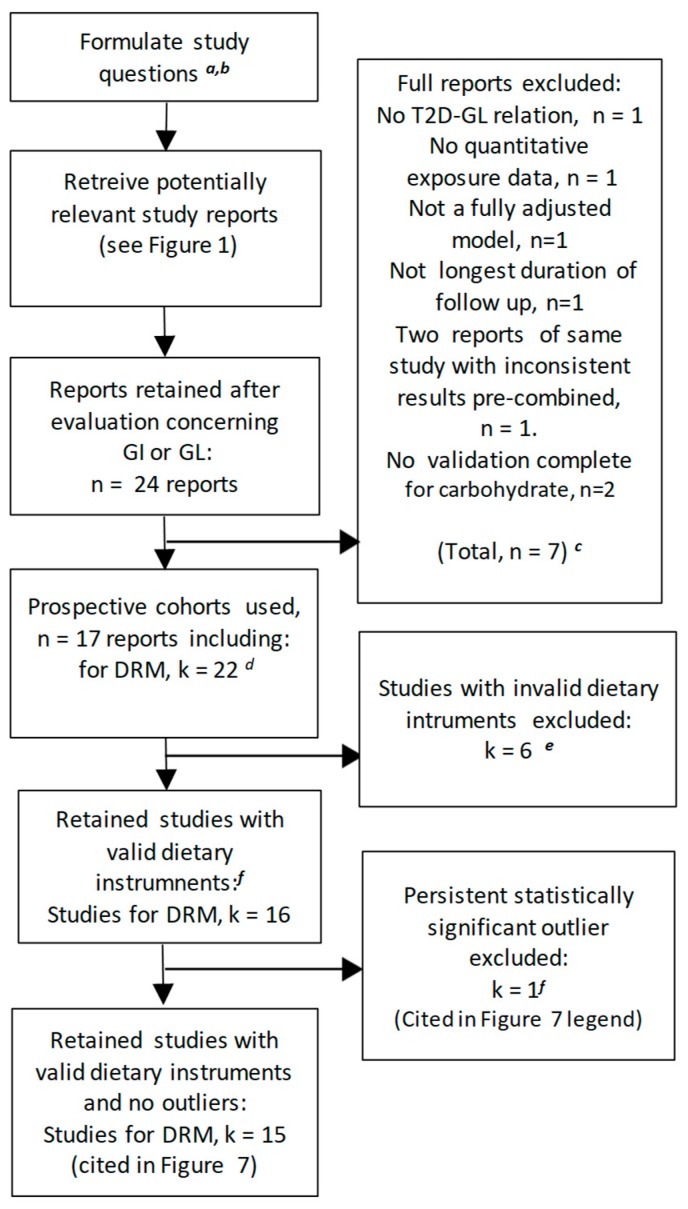
Process of the review and outcomes of the inclusion and exclusion criteria for the T2D–GL risk relation. Literature searches were for prospective cohort studies investigating this relation published in the period of 1946 to 6 December 2018 via MEDLINE and EMBASE and other sources—see the Methods section and Figure 1. Superscripts note: ***a***. Whether T2D is related to the dietary glycemic load (GL) and whether the risk relation is > 1.20 with a lower confidence limit > 1.10 for each 10th to the 90th percentile of GL intake (which, on analysis, was per 80 g GL in 2000 kcal (8400 kJ)? ***b***. Is the T2D–GL risk relation conditional on the validity of the dietary instrument for carbohydrate, on the sex of the participants, or on the BMI (kg/m^2^), and was the T2D–GI risk relation conditional on protein intake [10] and/or alcohol (ALC) intake [40]? ***c***. Details of the full reports excluded are described in the associated text and in more detail in Section 3 (h to l) of the Appendix A. ***d***. Studies used (n = 22) in the meta-analysis with multiple covariates. ***e***. Studies were invalid if the dietary instrument had a correlation coefficient for carbohydrate foods ≤ 0.55. ***f***. When a study suspected of being an outlier in a forest plot was proven to be so upon meta-regression analysis (*p* < 0.05) and persisted to be so in several different meta-analyses (Appendix A), it was excluded. Abbreviations: DRM, dose-response meta-analysis; GL, glycemic load; n, number of published reports; k, number of studies (more than one may appear in one report); T2D, type 2 diabetes.

**Figure 7 nutrients-11-01280-f007:**
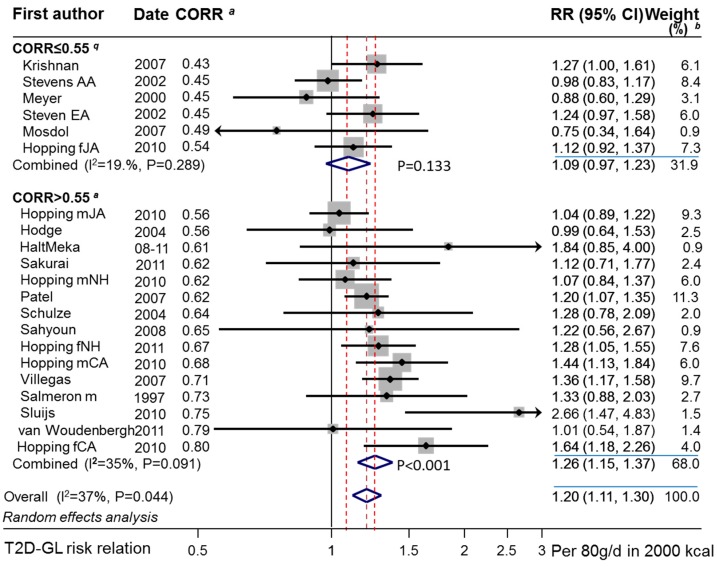
Forest plot of the dose-response T2D–GL risk relation by the validity of the dietary instrument. The box sizes are proportional to the weight contributed by the study to the combined-study mean. Horizontal lines span the individual study 95% confidence intervals. Arrowheads indicate truncations. Diamonds represent the combined-study mean RR values per 80 g GL in 2000 kcal diet (based on the glucose standard), and the corresponding 95% CI values. Analysis was undertaken on the natural logarithms and shown untransformed. *p*-values were calculated using the z test for RR and the Chi^2^ test for I^2^. Eligible studies were: Krishnan et al. (2007) [43], Stevens et al. (2002) [66] (two studies), Meyer et al. (2000) [64], Mosdol et al. (2007) [65], Hopping et al. (2010) [42] (six studies), Hodge et al. (2004) [69], Mekary et al. (2011) [40], Halton et al. (2008) [4] pre-combined as “HaltMeka”, Sakurai et al. (2012) [71], Patel et al. (2007), [61], Schulze et al. (2004) [45], Villegas et al. (2007) [46], Salmeron et al. (1997) [60] in men, and Sluijs et al. (2010) [62]. For studies with CORR > 0.55 and considering the relation with CORR (*p* = 0.002), the study of Simila et al. (2011) [41] was an outlier (*p* = 0.013) whereas the study of Sluijs et al. (2010) was not (*p* = 0.052). Abbreviations: AA, African-American; EA, European-American; CI, confidence interval; HPS, Health Professionals’ Study; I^2^; inconsistency; NHS, Nurses’ Health Study; *p*, probability; RR, relative risk. Superscripts: ***a***, CORR refers to the dietary instrument’s correlation coefficient for carbohydrate or carbohydrate foods; ***b***, percentage weights are from random effects.

**Figure 8 nutrients-11-01280-f008:**
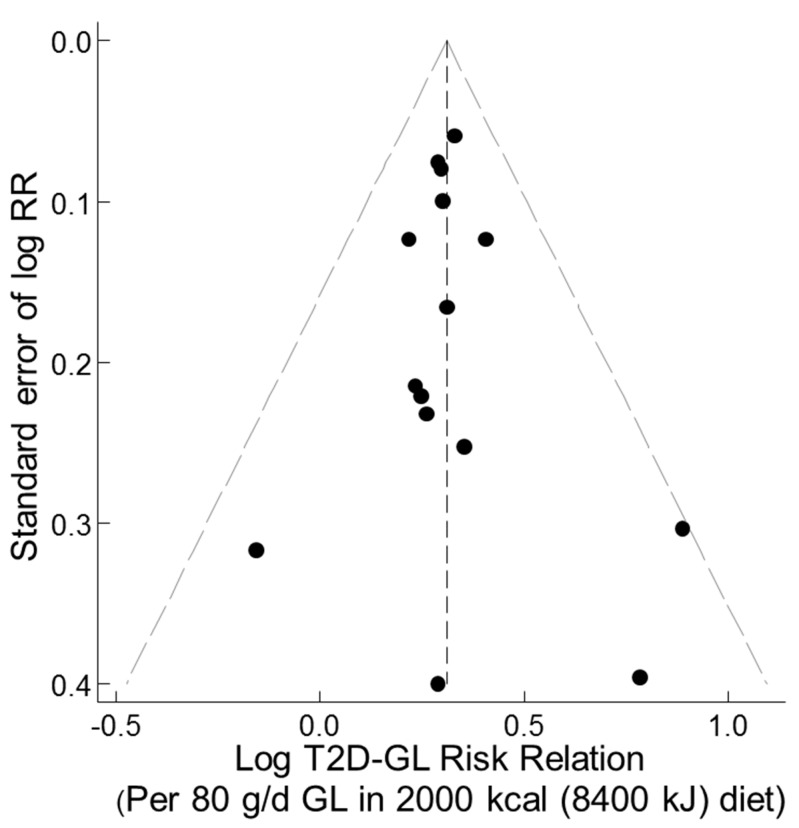
Funnel plot of the study results for the T2D–GL risk relation for valid studies. Valid studies were those with a correlation coefficient for carbohydrate or carbohydrate foods (CORR > 0.55, deemed a priori) and not including one outlier study [41] (Table 4 footnote *g*) (*p* < 0.05, deemed a priori) in the random effects model. Observations were adjusted for variation in CORR > 0.55 since this was a significant covariate (*p* < 0.001) deemed to be taken into control a priori [13]. The combined studies’ random effects trim-and-fill mean, when exponentiated, was 1.36 (1.28–1.45) (*p* < 0.001, n = 15 studies) per 80 g/d GL in a 2000 kcal (8400 kJ) diet. Trim-and-fill analysis identified missing studies = 0 to obtain a symmetrical distribution of observations. Adjustment for CORR was centered on a correlation coefficient of 0.7 (deemed a priori).

**Figure 9 nutrients-11-01280-f009:**
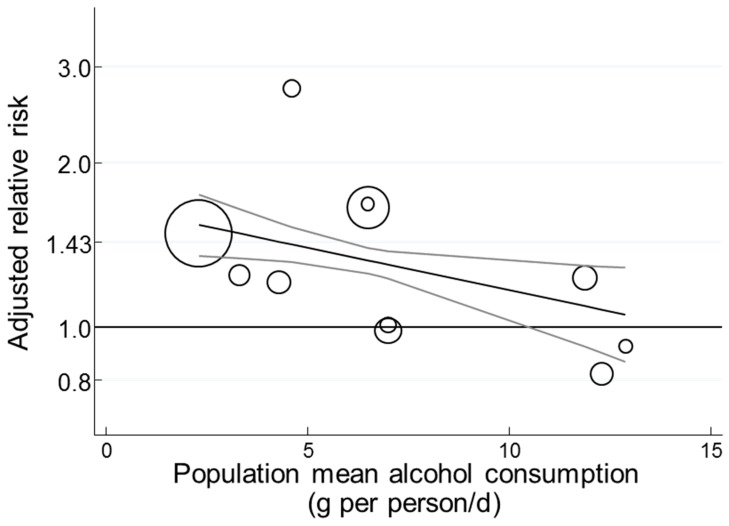
Possible dependence of the size of the T2D–GL risk-relation on the average population ALC consumption. The T2D–GL risk relation shown is per 80 g/d GL in a 2000 kcal (8400 kJ) diet. The negative slope was significant (*p* = 0.039, n = 11 studies). Circles represent individual study relative risks and have a size proportional to the study weight (larger circles: Greater weight). Lines show the trend and 95% CIs. Risk relations were adjusted for covariates (CORR, ethnicity (ETH), and duration of follow-up in years (FUY) defined in Section 3.3.14) while the population mean ALC intake was centered on 7g/d: These intakes were: Villegas et al. (2007) (2.3 g/d) [46], Schulze et al. (2003) (3.3 g/d) [45], Hodge et al. (2004) (4.3 g/d) [69], Sluijs et al. (2010) (4.6 g/d) [62], Halton et al. (2008) at 20-year follow-up [4] pre-combined with Mekary et al. (2011) at 26-year follow-up from the same study [40] without increasing the weight of the study (6.5 g/d), Krishnan et al. (2007) (6.5 g/d) [43], van Woudenbergh et al. (2011) (7 g/d) [43], Meyer et al. (2000) (7 g/d) [64], Salmeron et al. (1997) in men (11.9 g/d) [60], Similar et al. (2011) (11.9 g/d) [41], and Mosdol et al. (12.9 g/d) [65].

**Figure 10 nutrients-11-01280-f010:**
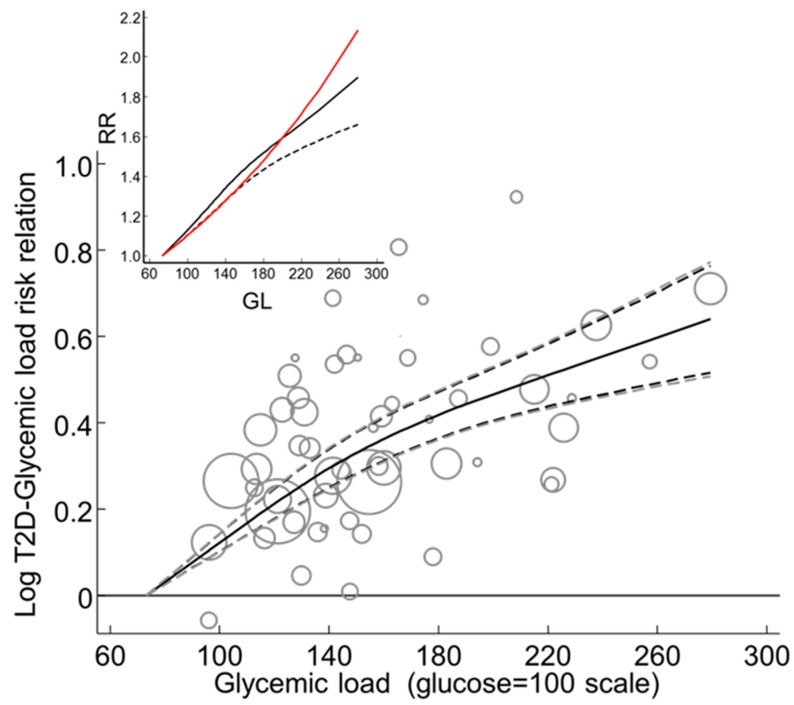
Global dose response meta-analysis of the T2D–glycemic load risk relation. The larger plot shows the mean increment in log risk relation and 95% CI values. Curves are the mean cubic-spline log risk relation (black, continuous) and the 95% CI values based on random effects analysis. The inner (black dashed) shows the conventional CI values based on (τ^2^ + σ^2^) whereas the outer (grey dashed) were based on an additional, forecasting term, σ_f_^2^, thus (τ^2^ + σ^2^ + σ_f_^2^), so that the 95% CIs were wider. **Insert:** Shows the RR (black, continuous line), the lower 95% confidence limit accounting for the three sources of error (black dashed line), and the corresponding log-linear prediction (red). The risk relation (RR) was per 80 g/d GL in a 2000 kcal (8400 kJ) diet.

**Table 1 nutrients-11-01280-t001:** Type 2 diabetes–glycemic index risk relations combined by a meta-analysis of the results of published prospective cohort studies using two steps: (i) Dose-response analysis and (ii) meta-analysis of the dose response results.

	Number of Studies	Model	Mean Relative Risks (95% CI)		*p*-Value for RR	Inconsistency (I^2^%)	Heterogeneity (τ ^2^)	*p*-Value for τ^2^ and I^2^
			Per 10 Units GI *^a^*					
(1) Studies with CORR ≤ 0.55:								
All such eligible studies ***^b^***	7	Fixed	1.02	(0.98–1.06)	0.377	-	-	-
		Random	1.05	(0.97–1.14)	0.271	65	0.007	0.009
(2) Studies with CORR > 0.55:								
All such eligible studies ***^c^***	10	Fixed	1.26	(1.21–1.31)	<0.001	(0) ***^d^***		-
		Random	1.27	(1.15–1.40)	<0.001	68	0.013	<0.001
Men-only and	6	Fixed	1.26	(1.21–1.31)	<0.001	(0)		-
women-only combined ***^e^***		Random	1.29	(1.15–1.45)	<0.001	79	0.014	<0.001
Women-only	3	Fixed	1.25	(1.20–1.30)	<0.001	(0)		-
studies ***^f^***		Random	1.29	(1.10–1.50)	<0.001	90	0.017	<0.001
Men-only	3	Fixed	1.32	(1.16–1.51)	<0.001	(0)		-
Studies ***^g^***		Random	1.31	(1.06–1.63)	0.013	47	0.017	0.15
Largest three studies	3	Fixed	1.25	(1.20–1.30)	<0.001	(0)		
(China and the USA) ***^h^***		Random	1.29	(1.11–1.50)	0.002	90	0.017	<0.001

^***a***^ The combined mean sample population range of GI values (on the glucose scale) across quintiles was 10.2 units GI. ^***b***^ Studies with correlation coefficients for carbohydrate <0.55 were: Krishnan et al. (2007) [43], Meyer et al. (2000) [64], Mosdol et al. (2007) [65], Oba et al. (2013) (in women) [44], Stevens et al. (2002) (two studies) [66], and Sluijs et al. (2013) [67], which included regions with correlation coefficients < 0.55. ^***c***^ Studies were: Barclay et al. (2007) [68], Bhupathiraju et al. (2014) (2 studies: NHS II, HPFS) [10], Hodge et al. (2004) [69], Mekary et al. (2011)(NHS I) [40], Oba et al. (2013) in men [44], Sahyoun et al. (2008) [70], Sakurai et al. (2012) [71], van Woudenbergh et al. (2011) [72], and Villegas et al. (2007) [46]. The study of Simila et al. (2002) [41]) was an outlier (*p* = 0.033) and was excluded. ^***d***^ All such I^2^ in brackets rely on fixed effects analysis, which presumes the true I^2^ to be zero, which some authors have favored over the underweighting of large studies relative to smaller studies in random effects meta-analysis [10,49,52]. All other analyses are random effects except where I^2^ = zero. ^***e***^ Men-only and women-only studies combined: Women-only studies were those at footnote *f*. Men-only studies were those at footnote *g*. In the meta-analysis of men only and women-only studies combined, the study of Simila et al. (2011) [41] was an outlier (*p* = 0.029). ^***f***^ Women-only studies were: Bhupathiraju et al. (2014) (NHS II) [10], Mekary et al. (2011) (NHS I) [40], and Villegas et al. (2007) [46]. ^***g***^ Men-only studies were: Bhupathiraju et al. (2014) (HPFS) [10], Sakurai et al. (2012) [71], and Oba et al. (2013) [44], excluding the significant outlier study of Simila et al. (2011) [41] (*p* = 0.049). ^***h***^ The three largest studies by population sample size were: Villegas et al. (2007) [46] (63,727 persons), Mekary et al. (2011) (NHS I) [40] (75,457 persons), and Bhupathiraju et al. (2014) (NHS II) [10] (83,540 persons). Abbreviations: CI, confidence interval; CORR, the food instruments’ correlation coefficient for carbohydrate; *p*, probability; RR, relative risk; I^2^, inconsistency, which is the among-studies variance (τ2) as a percentage of the sum of the among- and within-studies variances.

**Table 2 nutrients-11-01280-t002:** The type 2 diabetes–glycemic index dose-response risk relation in studies making study-level adjustments for specific nutrients. Studies with CORR > 0.55 were included in this analysis.

Study-Level Adjustment Made within Studies		Number of Studies	Model *^a^*	Mean Combined Relative Risk and (95%CI)		p-Value for RR	Inconsistency I^2^	Heterogeneity τ ^2^	*p*-Value for τ ^2^ and I^2^
				Per 10 units GI			(%)	*(Per 10*	
								*units GI)^2^*	
All studies with CORR > 0.55 bar one outlier [40] ***^b^***		10	Random	1.27	(1.15–1.40)	<0.001	68	0.013	<0.001
All fiber types ***^c^***		7	Random	131	(1.17–1.47)	<0.001	76	0.0138	<0.001
Cereal fiber ***^d^***		3	Random	1.39	(1.26–1.53)	<0.001	45	0.0034	0.161
Vegetable fiber ***^e^***		1	Random	1.50	(0.95–2.36)	0.080	-	-	-
Magnesium ***^f^***		2	Random	1.28	(0.94–1.79)	<0.001	69	0.0355	0.070
Protein ***^g^***		3	Random	1.39	(1.26–1.53)	<0.001	45	0.0034	0.161
Red meat ***^h^***		1	Random	1.47	(1.35–1.60)	<0.001	-	-	-
Alcohol ***^i^***		9	Random	1.26	(1.13–1.40)	<0.001	71	0.0132	<0.001
Energy ***^j^***		8	Random	1.27	(1.14–1.40)	<0.001	74	0.0135	<0.001
Saturated fats ***^k^***		3	Random	1.28	(1.07–1.48)	0.007	53	0.0110	0.012
Trans fats *^l^*		3	Random	1.39	(1.26–1.53)	<0.001	45	0.0034	0.161

^***a***^ Model procedures: (i) Analysis of doses response, (ii) random effects meta-analysis of the dose-response results. ^***b***^ Values from Table 1. Studies were: Barclay et al. (2007) [68], Bhupathiraju et al. (2014) (2 studies: NHS II, HPFS) [10], Hodge et al. (2004) [69], Mekary et al. (2011) (NHS I) [40], Oba et al. (2013) (men) [44], Sahyoun et al. (2008) [70], Sakurai et al. (2012) [71], van Woudenbergh et al. (2011) [72], and Villegas et al. (2007) [46]. The study of Simila et al. (2012) [41]) was an outlier (*p* = 0.033) (Table 1, footnote *c*). ^***c***^ Combined from Villegas et al. (2010) [46], Sakurai et al. (2011) [71], Oba et al. (2013) in men [44], Mekary et al. (2011) [40], Bhupathiraju et al. (2014) (2 studies; NHS I and HPFS) [10], and Barclay et al. (2007) [68]. ^***d***^ Combined from: Mekary et al. (2011) [40] and Bhupathiraju et al. (2014) (2 studies, NHS II and HPS) 10]. ^***e***^ Study was that of Barclay et al. (2007) [68]. ^***f***^ Combined from Oba et al. (2013) for men [44] and Schulze et al. (2004) [45]. ^***g***^ Combined from those studies cited in footnote *d*. ^***h***^ Mekary et al. (2011) [40]. ^***i***^ Combined studies of footnote *b* except Barclay et al. [68]. ^***j***^ Combined studies of footnote *b* except Barclay et al. [68] and Sahyoun et al. (2008) [70]. ^***k***^ Combined studies from Bhupathiraju et al. (2014) (2 studies, NHS II and HPS) [8] and van Woudenbergh et al. (2011) [72]. l Combined from: Mekary et al. (2011) [40] and Bhupathiraju et al. (2014) (2 studies, NHS II and HPS) [8]. Abbreviations: CI confidence interval; HPS, Health Professionals’ Study; NHS, Nurses’ Health Study; RR, relative risk; I^2^, inconsistency among studies, which is the ratio of the among-studies variance (τ ^2^) to the sum of the among-studies and within-studies variances expressed as a percentage.

**Table 3 nutrients-11-01280-t003:** Dose-response T2D–glycemic index risk relations by study region or ethnicity or ancestry for CORR > 0.55.

Region	Number of Studies	Sum Study Weights	Mean Risk Relation		*p*-Value for RR	I^2^	τ ^2^	*p*-Value for I^2^ and τ ^2^
			Mean	(95% CI)				
		(%)	Per 10 units GI/d			(%)		
1. Asia, east ***^a^***	3	33	1.17	(1.08–1.28)	<0.001	9	0.002	0.33
2. Australia ***^b^***	2	14	1.29	(1.05–1.58)	0.017	0	0.000	0.44
3. Europe ***^c^***	1	5	0.92	(0.63–1.35)	-	-	-	-
4. Euro-American ***^d^***	4	49	1.38	(1.24–1.52)	<0.001	43	0.005	0.16
5. European ancestry ***^e^***	7	68	1.32	(1.19–1.47)	<0.001	45	0.007	0.093
6. Eastern ancestry ***^f^***	3	33	1.27	(1.27–1.34)	<0.001	0	0.000	0.84
7. Eastern ancestry ^***g***^	4	--	1.26	(1.20–1.33)	<0.001	0	0.000	0.80

^***a***^ East Asian studies were: Oba et al. in men [30], Sakurai et al. [71], and Villegas et al. (2007) [46]. ^***b***^ Australian studies were: Barclay et al. (2007) [9] and Hodge et al. (2004) [69]. ^***c***^ Just one European study qualified and was van Woudenbergh et al. (2011) [72]. Although van Woudenbergh [63] was the only study validated as having CORR > 0.55 (and used clinical ascertainment of diabetes and adjusted for a family history of diabetes), it was an outlier (*p* < 0.012) among other studies meeting these criteria (Table 7
*n* = 5 studies, footnote *d*). The study of Simila et al. (2011) [41] was an outlier in multiple meta-analyses (Appendix A) and provided divergent results in [75], therefore, it was not included. Sluijs et al. (2013) [67] investigated eight European regions providing quantitative dose information on the combined studies only—some regions had dietary instruments with validity correlations < 0.55 while the combined studies’ validity correlation was not published; thus, the study was excluded. ^***d***^ Studies on European-Americans were: Bhupathiraju et al. (2014) (HPS and NHSII) [11], Mekary et al. (2011) [40], and Sahyoun et al. (2008) [70]. ^***e***^ Combined studies were from Australia, Europe, and European-Americans (the USA). ^***f***^ Combined studies were those listed in footnote *a*. These results were after adjustment of the RR for CORR, family history of diabetes and ALC (Section 3.2.14). ^***g***^ As in footnote *f* with an additional East Asian study included with CORR ≤ 0.55 (was 0.46) from Oba et al. in women [44], which was possible because of the adjustment for CORR as a covariate. Abbreviations: CI, confidence interval; CORR, the validity correlation for carbohydrate; I^2^, inconsistency; NHS, Nurses’ Health Study; HPS, Health Professionals’ Study; Pctl, percentile. RR; risk relation; τ ^2^, heterogeneity; T2D, type 2 diabetes.

**Table 4 nutrients-11-01280-t004:** The T2D–diabetes glycemic load risk relations combined with the dose-response meta-analysis of the results from published prospective cohort studies.

	Number of Studies	Model *^a^*	Mean Relative Risks (95% CI)		*p*-Value for RR	Inconsist-ency	Heterogeneity	*p*-Value for τ ^2^ and I^2^
			Per 80 Units GL per 2000 kcal *^b^*			(I^2^ %)	(τ ^2^)	
			(1) Studies with CORR ≤ 0.55					
1. All such eligible studies ***^c^***	6	Fixed	1.09	(0.99–1.21)	0.086	(0) ***^d^***	-	-
		Random	1.09	(0.97–1.23)	0.133	19	0.0040	0.289
			(2) Studies with CORR > 0.55					
2. All such eligible studies ***^e^***	15	Fixed	1.24	(1.16–1.32)	<0.001	(0)		-
		Random	1.26	(1.15–1.37)	<0.001	35	0.0089	0.091
3. Clinical report of T2D in ≥0.97 of studies ***^f^***	9	Fixed	1.24	(1.14–1.36)	<0.001	(0)		
		Random	1.33	(1.14–1.55)	<0.001	58	0.0278	0.014
4. Adjusted for CORR (centered on 0.7) *^**g**^*	15	Fixed	1.36	(1.26–1.48)	<0.001	(0)	-	-
		Random	1.36	(1.26–1.48)	<0.001	0	0	0.78
5. Adjusted for CORR (centered on 0.7) and FHD (centered on 0.50) ***^h^***	21	Fixed	1.34	(1.24–1.46)	<0.001	(0)	-	-
		Random	1.34	(1.24–1.46)	<0.001	1	–0.0002	0.643
6. After 5. Women-only studies ***^i^***	8	Fixed	1.38	(1.27–1.51)	<0.001	(0)	-	-
		Random	1.38	(1.27–1.51)	<0.001	0	0	0.784
7. After 5. Men-only studies, *^**j**^*	5	Fixed	1.30	(1.16–1.44)	<0.001	(0)	-	-
		Random	1.30	(1.16–1.44)	<0.001	0	0	0.640
8. Three largest studies (China and the USA) ***^k^***	3	Fixed	1.29	(1.19–1.39)	<0.001	(0)		
		Random	1.29	(1.19–1.39)	<0.001	(0)	0	0.706

^***a***^ Meta-analyses used either the fixed effects inverse variance model of Mantel-Haenszel [50] or the random effects inverse variance model of DerSimonian and Laird [51]. ^***b***^ The mean sampled population range of GL was 80.9 g GL per 2000 kcal (8400 kJ)/d from the 10th the 90th percentile of GL intake. ^***c***^ The six studies were: Hopping et al. (2005) (Japanese-American Women) [42], Krishnan et al. (2007) [43], Meyer et al. (2000) [64], Mosdol et al. (2007) [65], and Stevens et al. (2002) (2 studies) [66]. ^***d***^ All I^2^ values in brackets rely on fixed effects analysis, which presumes the true I^2^ to be zero. ^***e***^ The 15 studies were: Hodge et al. (2004) [69], Hopping et al. (2010) (5 studies not FJA) [42], Mekary et al. (2011) for NHS I at the 26-year follow-up [40] pre-combined with Halton et al. (2008) for NHS I at the 20-year follow-up [4] due to their heterogeneous results, Patel et al. (2007) [61], Sahyoun et al. (2008) [70], Sakurai et al. (2012) [71], Salmeron et al. (1997) in men [60], Schulze et al. (2004) [45], Sluijs et al. (2010) [62], van Woudenbergh et al. (2011) [72], and Villegas et al. (2007) [46]. Simila et al. (2011) [41] was an outlier (*p* = 0.013). f The nine studies were from: Hopping et al. (2005) (5 studies) [42], Mekary et al. (2012) NHS I at the 26-year follow-up [40] pre-combined with Halton et al. (2008) NHS I [4] due to inconsistent results between reports from the same study (I^2^ = 95%), Salmeron et al. (1997) in men [60] and Schulze et al. (2004) [45], and Sluijs et al. (2010) [62]. Simila was excluded as an outlier (*p* = 0.042 and from multiple perspectives (Appendix A)). ^***g***^ The 15 studies were cited in footnote ^e^. CORR was a significant covariate (*p* < 0.001). The study of Simila was an outlier (*p* = 0.010). ^***h***^ The 21 studies were: Hodge et al. (2004) [69], Hopping et al. (2010) (6 studies) [42], Krishnan et al. (2007) [43], Mekary et al. (2011) for NHS I at the 26-year follow-up [40] pre-combined with Halton et al. (2008) for NHS I at the 20-year follow-up [4] due to these two studies reporting highly inconsistent results (I^2^ = 95%), Meyer et al. (2000) [64], Mosdol et al. (2007) [65], Patel et al. (2007) [61], Sahyoun et al. (2008) [70], Sakurai et al. (2012) [71], Salmeron et al. (1997) in men [60], Schulze et al. (2004) [45], Sluijs et al. (2010) [62], Stevens et al. (2002) (2 studies [66], van Woudenbergh et al. (2011) [72], and Villegas et al. (2007) [46]. The study of Simila et al. [37] was an outlier (*p* = 0.021). ^***i***^ The difference between women-only and men-only studies was not significant (*p* = 0.571). Values are after adjustment for covariates of CORR and FHD of n = 21 studies. The eight studies included Hopping et al. (2010) (3 studies) [42], Krishnan et al. (2007) [43], Mekary et al. (2011) for NHS I at the 26-year follow-up [40] pre-combined with Halton et al. (2008) for NHS I at the 20-year follow-up [4] due to heterogeneous results; Meyer et al. (2000) [64], Schulze et al. (2004) [45], Villegas et al. (2007) [46]. ^***j***^ Values are after adjustment for covariates of CORR and FHD of n = 21 studies. Studies included were Hopping et al. (2010) (3 studies) [42], Sakurai et al. (2012) [71], Salmeron et al. (1997) in men [60]. The study of Simila et al. (2011) [41] was an outlier (*p* = 0.034). ^***k***^ The three largest studies by population sample size were: Villegas et al. (2007) [46] (64,227persons), Mekary et al. (2011) (NHS I) [40] (81,827 persons), and Schulze et al. (2004) [45] (91,249 persons). Abbreviations: CI confidence interval; CORR, the dietary instrument validity correlation; FHD, family history of diabetes; GL, glycemic load; RR, risk relation; I^2^, inconsistency, which is the ratio of the among-studies variance to the sum of the among-studies and within-studies variances.

**Table 5 nutrients-11-01280-t005:** Type 2 diabetes–glycemic load risk relations combined by a meta-analysis of the results of published prospective cohort studies, with model adjustments for covariates of SEX, CORR, ETH, and FUY except when specified otherwise in the first column and/or footnote ***^a^***.

Subjects/Study		Number of Studies	Model	Mean Risk Relation			*p*-Value for RR	Inconsistency (I^2^)	*p*-Value for I^2^
				By Extreme	By Dose (GL)				
				Quintiles	Response	(95%CI)			
				Per 10th to	Per 80 g	Corresponding		*%*	
				90th pctl of	increment in	units, see left			
				population	GL in a 2000				
				GL	kcal diet				
1.	All eligible studies	22	Random		1.33	(1.21–1.45)	<0.001	4	0.66
	with centered covariates ***^b^***								
2.	Women-only studies ***^c^***	8	Random		1.42	(1.30–1.54)	<0.001	0 *^**d**^*	0.53
	Men-only studies ***^e^***	6	Random		1.23	(1.11–1.37)	<0.001	0	0.53
3.	Women-only BMI.	4	Random	1.23	- ***^g^***	(0.99–1.50)	0.067	0	0.91
	<25 or <27 ***^f^***								
	Women-only BMI	4	Random	1.33	- ***^g^***	(1.16–1.54)	<0.001	0	0.48
	≥ 25 or ≥27 ***^f^***								
4.	Additional adjustment for	22	Random		1.32	(1.20–1.44)	<0.001	3	0.41
	conditional protein intake ***^h^***								
5.	Replacement of	22	Random		1.31	(1.19–1.44)	<0.001	3	0.42
	SEX as covariate by								
	ALC as covariate								
	when known (SEX not kept) ***^i^***								
6.	Additional adjustment	22	Random		1.31	(1.20–1.44)	<0.001	2	0.50
	for ALC intake when								
	known (covariate SEX kept) ***^j^***								

^***a***^ Study results were adjusted for sex (centered on 50% male and 50% female), ethnicity (centered on 50% European-American and 50% others), the dietary instrument (usually FFQ) correlation for carbohydrate (centered on 0.7), and the number of follow-up years (centered on 10 years) [13] except where specified otherwise in footnotes *d*, *e*, *g*, *h*, and *i*. ^***b***^ The 22 studies were: Hodge et al. (2004) [69], Hopping et al. (2010) (6 studies) [42], Krishnan et al. (2007) [43], Mekary et al. (2011) for NHS I at the 26-year follow-up [40] pre-combined with Halton et al. (2008) for NHS I at the 20-year follow-up [4] due to these two studies reporting highly inconsistent results (I^2^ = 95%), Meyer et al. (2000) [64], Mosdol et al. (2007) [65], Patel et al. (2007) [61], Sahyoun et al. (2008) [70], Sakurai et al. (2012) [71], Salmeron et al. (1997) in men [60], Schulze et al. (2004) [45], Simila et al. (2011) [41], Sluijs et al. (2010) [62], Stevens et al. (2002) (2 studies [66], van Woudenbergh et al. (2011) [72], and Villegas et al. (2007) [46]. ^***c***^ Women-only studies: Values are after adjustment for covariates on footnote *a* except for SEX. The combined mean RR value for women was significantly higher than for men (*p* = 0.030). Studies included Hopping et al. (2010) (3 studies) [42], Krishnan et al. (2007) [43], Mekary et al. (2011) for NHS I at the26-year follow-up [40] pre-combined with Halton et al. (2008) for NHS I at the 20-year follow-up [4] due to their heterogeneous results, Meyer et al. (2000) [64], Schulze et al. (2004) [45], and Villegas et al. (2007) [46]. Note that Meyer et al. (2000) [64] had a potential low outlier RR value (*p* = 0.06), but it was retained because it did not meet the exclusion criterion of *p* ≤ 0.05. ^***d***^ Although a random effects model was applied, the result was a fixed effect because I^2^ = 0%. ^***e***^ Men-only studies: Values are after adjustment for covariates shown in footnote *a* except for SEX. The combined mean RR value for men was significantly lower than for women (*p* = 0.030). Studies included were Hopping et al. (2010) (3 studies) [42], Sakurai et al. (2012) [71], Salmeron et al. (1997) in men [60], and Simila et al. (2011) [41]. Note that Simila et al. (2011) [41] was potentially a low outlier (*p* = 0.06) RR value, but was retained because it did not meet the exclusion criterion of *p* ≤ 0.05. ^***f***^ Women-only BMI: Included studies were Schulze et al. (2004) [45], Krishnan et al. (2007) [43], Villegas et al. (2007) [46], and Oba et al. (2013) [44]. Data on GL values by quantiles in the BMI sub-cohorts were not reported in the original studies, which precluded dose (GI)-response meta-analysis. ^***g***^ Studies did not report intakes for GL by quantiles, therefore, extreme quintile meta-analysis was used. ^***h***^ The model using the 22 studies of footnote *b*, adjusted for the covariates shown in footnote *a* and adjusted for protein intake when given in the original prospective cohort studies adjustments were made simultaneously for energy, fat, fiber (or cereal fiber), and protein intakes; this conditional adjustment for protein has been hypothesized to elevate the T2D–GL RR estimate (endnote b). The conditional adjustment for protein intake used indicator variables (1 for the conditional adjustment for protein and 0 otherwise). Studies making this conditional adjustment were: Sluijs et al. (2010) [62], van Woudenbergh et al. (2011) [72], and Halton et al. (2008) [4] pre-combined with Mekary et al. (2011) [40] (who adjusted for red meat rather than protein). ^***i***^ The model using all 22 studies listed in footnote *b*, adjusted for covariates as in footnote *a*, except the centered covariate for SEX was set to zero when the population average ALC intake was known, plus it was further adjusted using a centered covariate for the average population ALC intake when known and zero when unknown. ^***j***^ The model using the 22 studies listed in footnote *b*, adjusted for covariates as in footnote *a*, plus also adjusted for the studies’ population average ALC intakes (g/d). This covariate adjustment was applied when the ALC intake was known for a study and was presented to the analytical model as a centered covariate, otherwise it was presented as zero when unknown. Abbreviations: BMI, body mass index (kg/m^2^); CI, confidence interval; CORR, covariate for the dietary instrument correlation for carbohydrate; ETH, covariate for ethnicity; FUY, covariate for follow-up years; GL, glycemic load, RR, risk relation; SEX, covariate for sex of participants; I^2^, inconsistency among studies which is ratio of the among-studies variance to the sum of the among- and within-studies variances expressed as a percentage.

**Table 6 nutrients-11-01280-t006:** The T2D–glycemic load risk relation with and without the three health professionals’ studies (NHS I, NHS II, and HPFS) in multi-covariate meta-analysis ***^a^***.

Studies Included in the Meta-Analysis with Covariates *^a^*	Number of Studies	Relative Risk (RR) and 95%CI		*p*-Value for RR	Inconsistency (I^2^)	*p*-Value for I^2^
		Based on Random Effects				
		Mean	(95% CI)			
		Per 80 g increment			%	
		in GL in a 2000 kcal (8400 kJ diet)				
All included studies ***^a^***^,^***^b^***	22	1.33	(1.21–1.45)	<0.001	4	0.66
Dropping health professionals’ NHS I, NHS II, and HPS studies ***^a^***^,^***^c^***	19	1.33	(1.20–1.48)	<0.001	12	0.31
Dropping health professionals’ and outlier studies from Meyer et al. (2000) and Sluijs et al. (2010) ***^a^***^,^***^d^***	17	1.34	(1.22–1.47)	<0.001	0	0.87

^***a***^ Adjusted for covariates of SEX (centered on 50% male and 50% female), the food frequency questionnaire correlation for carbohydrate (CORR, entered on 0.7), ethnicity (ETH, centered on 50% European-American and 50% others), and follow-up years (FUY, centered on 10 years). ^***b***^ The 22 published studies were as described in Table 5 footnote *b*. ^***c***^ The 19 published studies were: Hodge et al. (2004) [69], Hopping et al. (2010) (6 studies) [42], Krishnan et al. (2007) [43], Meyer et al. (2000) [64], Mosdol et al. (2007) [65], Patel et al. (2007) [61], Sahyoun et al. (2008) [70], Sakurai et al. (2012) [71], Simila et al. (2011) [41], Sluijs et al. (2010) [62], Stevens et al. (2002) (2 studies) [66], van Woudenbergh et al. (2011) [72], and Villegas et al. (2007) [46]. ^***d***^ The 17 published studies were those 19 of footnote *c* after withdrawing two outliers: Meyer et al. (2000) [64] and Sluijs et al. (2010) [62] (*p* = 0.042 and 0.023, respectively). Abbreviations: CI, confidence interval; GL, glycemic load; HPS, Health Professionals’ Study; NHS I, Nurses’ Health Study I; NHS II, Nurses’ Health Study II; RR, relative risk; I^2^, ratio of the among-studies variance to the sum of the among-studies and within-studies variances.

**Table 7 nutrients-11-01280-t007:** Dose response type 2 diabetes–glycemic index and glycemic load risk relations and study-level validity.

Studies		*n*	Mean Risk Relation	95%CI	*p*-Value	I^2^	Person Years	Figure Table or Section	Model Effects
			*Per 10 units GI*			*%*	*Millions*		
**T2D–Glycemic index relative risk**									
1	Studies with CORR ≤ 0.55. ^***a***^	7	1.05	(0.97–1.14)	0.271	65	-	Table 1	Random
2	Studies with CORR > 0.55. ***^b^***	10	1.27	(1.15–1.40)	<0.001	68	5.0	Table 1	Random
3	As 2 + study-level adjustment for clinical report of T2D. ***^c^***	6	1.40	(1.29–1.51)	<0.001	14	4.2	3.2.3	Random
4	As 3 + study-level adjustment for FHD. ***^d^***	5	1.41	(1.32–1.51)	<0.001	5	3.8	3.2.13	Random
			*Per 80 g/d GL in 2000 kcal*						
**T2D–Glycemic load relative risk**			*(8400 kJ) diet*						
1	Studies with CORR ≤ 0.55. ^***e***^	6	1.09	(0.97–1.23)	0.133	19	-	Table 4	Random
2	Studies with CORR > 0.55. ^***f***^	15	1.26	(1.15–1.37)	<0.001	35	4.4	Table 4	Random
3	As 2 + study-level adjustment for clinical report of T2D. ***^g^***	9	1.33	(1.14–1.55)	<0.001	58	3.8	Table 4	Random
4	As 3 + study-level adjustment for FHD. ***^h^***	4	1.61 ***^h^***	(1.18–2.12)	0.003	31	3.1	3.3.16	Random

^***a***^ Studies with CORR ≤ 0.55 (invalid dietary instruments for carbohydrate) were those in Figure 3 and Table 1 panel 1. ^***b***^ Studies with CORR > 0.55 (valid dietary instruments for carbohydrate). Studies were those in Figure 3 and Table 1 panel 2. The study of Simila et al. was an outlier (*p* = 0.033). ^***c***^ Studies using clinical (valid) records for ascertainment of T2D (self -report of T2D contributed to <3% of persons) were Sakurai et al. [71], Sahyoun et al. [70], Barclay et al. [68], Bhupathiraju et al. (HPS) [10], Bhupathiraju et al. (NHS II) [10], and Mekary et al. (NHS) [40]. The studies of Simila et al. [41] and van Woudenbergh et al. [72] were outliers (*p* < 0.001 and *p* = 0.029, respectively). ^***d***^ Studies with study-level adjusted for FHD were: Sakurai et al. [45], Barclay et al. [68], Bhupathiraju et al. (HPS) [10], Bhupathiraju et al. (NHS II) [10], and Mekary et al. (NHS I) [40]. Studies of Simila et al. [41] and van Woudenbergh et al. [72] remained outliers (*p* = 0.012 and *p* = 0.011, respectively). ^***e***^ Studies were those with CORR ≤ 0.55 in Figure 7 and Table 4 row 1. ^***f***^ Studies with CORR > 0.55 (valid dietary instruments for carbohydrates). Studies were those as in Figure 7 and Table 4 row 2. Simila et al. was an outlier (*p* = 0.013). ^***g***^ Studies using clinical (valid) records for the ascertainment of T2D (self -report of T2D contributed to <3% of persons) were: Schulze (2004) [69], Salmeron males (1997) [74], Sluijs et al. (2010) [62], Hopping et al. (2010) (5 studies not including Japanese women with CORR < 0.55) [42], and the studies of Mekary et al. (2011) [40] and Halton et al. (2008) [4] pre-combined. ^***h***^ Studies making study-level adjustment for FHD were: Schulze (2004) [68], Salmeron (1997) males [74], Sluijs et al. (2010) [62], and the study (NHS I) results of Mekary et al. and Halton et al. pre-combined [4,40]. Note that dropping the pre-combined studies of Mekary et al. and Halton et al. [4,40] left three studies for which RR was 1.59 (1.07–2.35). Abbreviations: CI, confidence interval; CORR, correlation coefficient for carbohydrates when validating the dietary instrument; ETH, ethnicity (European-American versus other ethnicities); FDH, family history of diabetes; FUY, follow-up years; GI, glycemic index; GL, glycemic load; I^2^, inconsistency among study results; n, number of studies; RR, relative risk; SEX, sex of participants; T2D, type 2 diabetes.

**Table 8 nutrients-11-01280-t008:** Dose-response type 2 diabetes–glycemic index and –glycemic load risk relations with adjustments for covariates.

		*n*	Mean Risk Relation	95%CI	*p*-Value	I^2^	Person Years	Figure or Table or Section		ModelEffects
**T2D–glycemic index risk relation**			*Per 10 units GI*			*%*	*Millions*			
1	Meta-analysis-level adjusted for CORR (centered on 0.7) (includes CORR ≤ 0.55 and CORR > 0.55). ^***a***^	15	1.25	(1.12–1.41)	<0.001	67	6.6	_ ***^a^***	Random	
2	As 1 dropping invalid studies (CORR ≤ 0.55). ***^b^***	10	1.24	(1.11–1.37)	<0.001	57	5.0	– ***^b^***	Random	
3	As 1 + meta-analysis level adjusted for studies having	14	1.28	(1.23–1.34)	<0.001	0	6.5	3.2.13	Random	
	made study-level adjustments for FHD									
	(centered on 0.5). ***^c^***									
4	As 3 + covariate for ALC intake (centered on 7 g/d). *^**d**^*	15	1.26	(1.21–1.32)	<0.001	0	6.9	3.2.14	Random	

			*Per 80 g/d GL in a 2000 kcal*							
**T2D–glycemic load risk relation**			*(8400 kJ) diet*							
1	Meta-analysis-level adjusted for CORR (centered on 0.7) (includes CORR ≤ 0.55 and >0.55). ***^e^***	21	1.32	(1.22–1.43)	<0.001	1	6.3	3.3.16	Random	
2	As 1 dropping invalid studies (CORR ≤ 0.55). ***^f^***	15	1.36	(1.26–1.48)	<0.001	0	4.4	3.3.5	Random	
3	As 1 + meta-analysis level adjustment for FHD (centered on 0.5). *^**g**^*	21	1.34	(1.24–1.46)	<0.001	0	6.3	Table 4	Random	
4	As 3 + covariate for population average ALC	21	1.35 ***^d^***	(1.22–1.49)	<0.001	0	6.3	3.3.16	Random	
	consumption (centered on 7 g/d). ***^h^***									

^***a***^ Studies were those in Figure 3 (both CORR > 0.55 and CORR ≤ 0.55, where known, which excluded Sluijs (2013) [67]). The study of Simila et al. [41] was an outlier from multiple perspectives (*p* = 0.042). The T2D–GI risk relation co-varied with CORR as nonsignificant (*p* = 0.204). Significance for CORR became apparent when FHD was a covariate (footnotes *c* and *d* and Section 3.3.16). The study of Sluijs (2013) [67] had no value for CORR for all regions combined in their multiregional study and so was not includable. ^***b***^ Studies were the same as those in Figure 3 for the CORR > 0.55. The study of Simila et al. [41] was an outlier (*p* = 0.049, Table 1 footnote *g*). The T2D–GI risk relation co-varied with CORR non-significantly (*p* < 0.28). ^***c***^ Studies adjusting for FHD at the study-level were Sakurai et al. [71], Barclay et al. [68], Krishnan et al. [43], Oba et al. (males) [44], Oba et al. (females) [44], Hodge et al. [69], Bhupathiraju (HPS) [10], Mekary (NHS I) [40], and Bhupathiraju (NHS II) [10]. The T2D–GI risk relation co-varied with CORR (*p* < 0.001) and FHD (*p* < 0.001). Studies in [41] and [72] were outliers (*p* = 0.009 and *p* = 0.015, respectively). When the covariate for FHD was centered on zero rather than 0.5, the RR was 1.54 (1.41–1.70) (Section 3.2.13), consistent with studies making study-level adjustments for FHD having higher RR values albeit non-significantly (*p* = 0.149), but plausible given the significance for GI (Table 7). ^***d***^ Studies were the same as those in the corresponding forest plot (Figure 3) (both CORR ≤ 0.55 and CORR > 0.55, where known, but excluded Sluijs (2013) [67]) for which CORR was unknown. Studies’ average population ALC intakes were (g/d): Mosdol (12.9) [65], Krishnan (6.5) [43], Simila (11.3) [42], Oba (males) (29.7) [44], Hodge (4.3) [34], Bhupathiraju (HPFS) (11) [10], Meyer (7) [64], Oba (females) (1.9) [44], Villegas (2.3) [46], Mekary (6.2) [22], and Bhupathiraju (NHS II) (3) [10]. The T2D–GI risk relation co-varied with CORR, FHD, and population average ALC consumption (*p* < 0.001, *p* < 0.001, and *p* = 0.007, respectively). The study of Simila et al. [41] was inlying due to the moderately higher ALC consumption (11 g/d). The covariate for the population average ALC consumption was significant (*p* = 0.007). The study in [72] remained an outlier (*p* = 0.012). The study in [67] was excluded because the combined regional study did not co-report a combined regional correlation coefficient for carbohydrates, while the results for each regional study were not supplied with quantitative information on GI intakes. The covariate was significant for CORR (*p* < 0.001), FHD (*p* < 0.001), and the population average ALC consumption (*p* = 0.007). ^***e***^ Studies were the same as those listed in Figure 7 (both CORR > 0.55 and CORR ≤ 0.55). The T2D–GL risk relation co-varied with CORR (*p* < 0.001). The study of Simila et al. [41] was an outlier (*p* = 0.016). ^***f***^ Studies were the same as those listed in Figure 7 for CORR > 0.55 only. The T2D–GL risk relation co-varied with CORR (*p* < 0.001). The study [41]] was an outlier (*p* = 0.010). ^***g***^ Studies were the same as those listed in Figure 7 (both CORR > 0.55 and CORR ≤ 0.55). The T2D–GL risk relation co-varied with CORR (*p* < 0.001), but not FHD (*p* = 0.149). The study in [41] was an outlier (*p* = 0.021). When the covariate for FHD was centered on zero rather than 0.5, the RR was 1.40 (1.25–1.56) (Section 3.3.16), consistent with studies making study-level adjustments for FHD with higher RR values albeit non-significantly (*p* = 0.149), but plausible given the significance for GI (Table 7 row 3 for GI). ^***h***^ Studies were those listed in Figure 7 (both CORR ≤ 0.55 and CORR > 0.55). The T2D–GL risk relation co-varied with CORR (*p* = 0.001), but not FHD (*p* = 0.232) or the average population ALC consumption (*p* = 0.989). The study of Simila et al. [72] remained an outlier (*p* = 0.037) despite ALC (centered on 7 g/d) being a covariate. Inclusion of the study of Simila et al. [41] and dropping the covariate for FHD returned some significance to ALC as a covariate (*p* = 0.09), but truly so only when ethnicity (ETH, European-American vs. all other ethnicities as in [13]) was a covariate, when ALC was significant as a covariate (*p* = 0.040) as in Figure 9. Abbreviations: CI, confidence interval; CORR, correlation coefficient for carbohydrate when validating the dietary instrument; ETH, ethnicity (European-American versus other ethnicities); FDH, family history of diabetes; FUY, follow-up years; GI, glycemic index; GL, glycemic load; I^2^, inconsistency among study results; n, number of studies; RR, risk relation, T2D, type 2 diabetes.

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
