# Peer review of "Dietary Glycemic Index and Load and the Risk of Type 2 Diabetes: A Systematic Review and Updated Meta-Analyses of Prospective Cohort Studies"

_nutrients, 2019, doi:10.3390/nu11061280_

Reviewer 1 Report

This represents a solid well considered review of meta-analyses on the topic of GI and GL relative to the RR for developing T2DM...i have no substantive criticisms or remarks, other than the footnote 'a' on line 71 to GI and GL has not been included at the bottom of the page [presumably to denote definitions for the two measures in question, i.e.  Glycemic Index and Glycemic Load, which should indeed, be defined,].

Author Response

Responses to Reviewer 1

 Comments and Suggestions for Authors

This represents a solid well considered review of meta-analyses on the topic of GI and GL relative to the RR for developing T2DM...i have no substantive criticisms or remarks, other than the footnote 'a' on line 71 to GI and GL has not been included at the bottom of the page [presumably to denote definitions for the two measures in question, i.e.  Glycemic Index and Glycemic Load, which should indeed, be defined,].

Thank you for your favourable comments and the suggestion to use ‘footnotes’ (rather than ‘endnotes’), which was useful.

Because there is limited space in a footnote we used endnotes, which are placed after the Discussion section. Never the less, it was useful to use footnotes in the first instances, which highlight the content of the endnote and where the endnote can be found.

Because we refer to endnote b three times, we use a leading footnote for the first instance (page 6 near line 204) , then in-text parentheses for the second and third instances at currently near lines 270 and 976. We trust this is satisfactory.

Reviewer 2 Report

This was an extremely thorough meta-analysis of prospective cohort studies examining the relationships between dietary glycemic index (GI; 10 studies) and glycemic load (GL; 15 studies) and the risk of type 2 diabetes (T2D) among healthy adults.  The authors have concluded that there is increased risk for incident T2D of 27% per 10 units GI, and 26% per 80 g/d GL in a 2000 kcal diet.  They also reported statistically significant dose-responses for both GI and GL. The meta-analyses focused on analysis of studies according to whether they used truly valid dietary instruments (i.e., validity correlations for carbohydrate consumption >0.55), which is an aspect they describe as not having been considered in most previous meta-analyses on this topic. The authors have also conducted numerous subgroup and sensitivity analyses to examine the several suspected sources of heterogeneity, including (but not limited to) ascertainment of T2D, protein and alcohol intakes, sex, ethnicity, body mass index, family history of diabetes, number of dietary assessments and geographical region.  The paper is quite long, and many journals would require that some of the information be moved to a supplemental appendix, although this may not be necessary with the Nutrients format.  Please see below my specific comments.

Specific Comments to the Authors

1.     Abstract:  Lines 50-52, this meaning of this sentence is not clear, particularly the phrase, “It is over a decade ago that meta-analyses took into account…”

2.     Lines 83-85, this sentence is not clear.

3.     Line 96, suggest specifying the number of meta-analyses that you are referring to where you say “Among all these meta-analyses”

4.     Lines 100-102, suggest that you describe how the >0.55 validity correlation for carbohydrate was selected. Line 101, what is meant by CHD-GI, should this instead by T2D-GI?

5.     Lines 103-14, suggest that you provide the date in the text for that 1st meta-analysis.

6.     Line 117, suggest deleting or rewording the sentence “This effect is a double blow.”

7.     Line 195, “rations” should be ratios

8.     Section 2.1.5., suggest that the acronyms used in this section for family history of diabetes (FHD), alcohol (ALC), etc. can be used throughout the paper.

9.     Lines 217-219, instead of “sought” should this be “obtained”? Sought suggests that you asked the other authors for that information, but not necessarily that it was received from them.

10.  Line 233, “untaken” should be undertaken.

11.  Lines 280-282, this does not appear to be a complete thought, and it is confusing as written.

12.  Line 283, “hypothesize” should be hypothesized

13.  Lines 301-303, the statement about extreme quantile meta-analysis is not clear, specifically the factors of 1.067, 1.20 and 0 factors for median tertiles, median quartiles and median quintiles, respectively.

14.  Table 1 shows results for both fixed- and random-effects meta-analysis models.  A fixed-effect model requires that the true effect size does not vary from study to study, which is more restrictive and seems unlikely (implausible) from a theoretical perspective.  Under the random-effects model there is a distribution of true effects and the summary relative risk is an estimate of that distribution’s mean.  Accordingly, I recommend limiting the analyses to random effects models.

15.  Figure 1 suggest that it would be helpful to include boxes on the in-text study inclusion figure (for both GI and GL) in the main body of the paper. Although I understand it has been included as a supplemental figure, I think it would be helpful to show the starting number of studies with a categorization of why they were not included, instead of just starting the figure at 24 reports. The sentence about prior to 1997 is not clear.

16.  Line 465, “Sensitive” should be Sensitivity

17.  Line 470, what does “within” mean at the end of this sentence?

18.  Line 475, “ask” should be asked; Line 495, “instant” should be instance; Line 513, do you mean “follow” instead of “fall”?

19.   Line 775, I think “without statically inconsistent” should be “without being statistically significant”; Line 777, “publish” should be published

20.  Line 839, suggest explaining more the statement that “two studies included in the analysis in [15] have been questioned as includable, [11] others have used them similarly.”

21.  Line 843, “sensitive” should be sensitivity; Line 934, “assumption” should be assumptions; Line 969, “inatake” should be intake; Line 980 “adjusmnet” should be adjustment; Line 987, “instrument” should be instrument

22.  Line 1003, I think the sentence “Some effects nevertheless was not excludable and was plausible considering the significance for GI” needs to be reworded

23.  Lines 1010 and 1019, the use of the phrase “at present” is confusing; Line 1011, it is not clear what is meant by “Recall adjustments”

24.  Line 1042, I think here instead of T2D-GI you mean T2D-GL

25.  Lines 1074-1075, while it is true that you included studies that had valid dietary instruments for estimating exposures, you also conducted analyses where the CORR was<0.55. Suggest mentioning that here too.

26.  Lines 1080 and 1088, saying that the outcome was the T2D-GI relation which was our primary outcome for GI (or for GL in the second instance) is not clear.

27.  Line 1092, do you mean P<0.05 instead of P<0.50 for statistical significance?

28.  Line 1213, “ads” should be adds

29.  Lines 1250-1263, I think the public health relevance discussion is important. Could this be incorporated into the Abstract as well?

Author Response

Responses to Reviewer 2 .

This was an extremely thorough meta-analysis of prospective cohort studies examining the relationships between dietary glycemic index (GI; 10 studies) and glycemic load (GL; 15 studies) and the risk of type 2 diabetes (T2D) among healthy adults.  The authors have concluded that there is increased risk for incident T2D of 27% per 10 units GI, and 26% per 80 g/d GL in a 2000 kcal diet.  They also reported statistically significant dose-responses for both GI and GL. The meta-analyses focused on analysis of studies according to whether they used truly valid dietary instruments (i.e., validity correlations for carbohydrate consumption >0.55), which is an aspect they describe as not having been considered in most previous meta-analyses on this topic. The authors have also conducted numerous subgroup and sensitivity analyses to examine the several suspected sources of heterogeneity, including (but not limited to) ascertainment of T2D, protein and alcohol intakes, sex, ethnicity, body mass index, family history of diabetes, number of dietary assessments and geographical region.  The paper is quite long, and many journals would require that some of the information be moved to a supplemental appendix, although this may not be necessary with the Nutrients format.  Please see below my specific comments.

Thank you for this favourable summary,

Specific Comments to the Authors

1.     Abstract:  Lines 50-52, this meaning of this sentence is not clear, particularly the phrase, “It is over a decade ago that meta-analyses took into account…”

Thank you. We reworded as follows:

“It is now over a decade ago that a published meta-analysis used a pre-defined standard to identify valid studies. Considering valid studies only, and using random effects dose-response meta-analysis….”

Thak you we revised

2.     Lines 83-85, this sentence is not clear.

 Thank you.  We reworded as follows:

“Outcomes were published T2D-GI RR values which ranged from 1.12 to 1.45, and T2D-GL RR values that also ranged from 1.12 to 1.45. For these outcomes the exposures to GI and GL approximately spanned from the 10th to the 90th percentiles for typical population intakes GI and GL.”

3.     Line 96, suggest specifying the number of meta-analyses that you are referring to where you say “Among all these meta-analyses”

Thank you. We insert the references rather than the number of meta-analyses as this more clearly states the publications to which we refer,

Among all these meta-analyses  [6, 10, 11, 12, 13, 15]

4.     Lines 100-102, suggest that you describe how the >0.55 validity correlation for carbohydrate was selected. Line 101, what is meant by CHD-GI, should this instead by T2D-GI?

a) We now explain: .  “This higher value (0.55) was selected on the basis of the study of Brunner et al [18] who for nutrients in general recommended a value of 0.5, which was adopted by Barclay et al in 2008 [10] in their studies of T2D-GIand GL risk ratios, to which 10% was added allowing for error of estimation”

b) “CHD-GI and GL”  at this point  was expanded  to “coronary heart disease-GI and GL” to lose the uncertainty. (ie it was not the T2D-GI and GL   relationships)

5.     Lines 103-14, suggest that you provide the date in the text for that 1st meta-analysis.

Thank you, we inserted “, which was published in 2008” into the text.

6.     Line 117, suggest deleting or rewording the sentence “This effect is a double blow.”

Thank you. We replaced with “The effect is 2-fold”

7.     Line 195, “rations” should be ratios

Thank you. Corrected.

8.     Section 2.1.5., suggest that the acronyms used in this section for family history of diabetes (FHD), alcohol (ALC), etc. can be used throughout the paper.

Thank you, this has been done  for both ALC and FHD

9.     Lines 217-219, instead of “sought” should this be “obtained”? Sought suggests that you asked the other authors for that information, but not necessarily that it was received from them.

Thank you, Corrected.

10.  Line 233, “untaken” should be undertaken.

Thank you. Corrected..

11.  Lines 280-282, this does not appear to be a complete thought, and it is confusing as written.

Thank you..  Global dose response meta-analysis is already dealt with  in  Section 2.2.2.  The matter at 280-282  is unnecessary and incomplete, seemingly  accidental spurious paste. The matter has been removed.

12.  Line 283, “hypothesize” should be hypothesized

Thank you. Corrected.

13.  Lines 301-303, the statement about extreme quantile meta-analysis is not clear, specifically the factors of 1.067, 1.20 and 0 factors for median tertiles, median quartiles and median quintiles, respectively.

Thank you, this detail should have been deleted. It was legacy  information for a prior draft of our work in which  EQM results were more extensively reported.

14.  Table 1 shows results for both fixed- and random-effects meta-analysis models.  A fixed-effect model requires that the true effect size does not vary from study to study, which is more restrictive and seems unlikely (implausible) from a theoretical perspective.  Under the random-effects model there is a distribution of true effects and the summary relative risk is an estimate of that distribution’s mean.  Accordingly, I recommend limiting the analyses to random effects models.

Thank you for this account.

 From time to time this theoretical basis is unsatisfactory for some researchers [1], including some powerful statisticians [2]. In addition statistical software allows for the output of both random effects and fixed effects summary measures in the same forest plot.

Therefore rather than remove the fixed effects information we modified  Lines 295 to 296 in the original submission, which read:

  “Fixed effects was also applied for comparison with random effects [51, 52] [I2=(0)] a used elsewhere for the T2D-GI and GL relative risks [11] but was not a primary outcome.”

The sentence was replace by a paragraph (to help the information stand out better) which now reads:

“In Tables, fixed effects results are shown in addition to random effects results, on which there is discussion elsewhere applicable to intervention studies  [51, 52] [I2=(0)] and because some users have considered fixed effects more appropriate when the range of observational study sizes is large [11]; the random effects is then thought to under weigh the large studies. Thus a fixed effect alone was applied in one published meta-analysis of the T2D-GI and GL relative risks [11]. However, random effects have a more generally accepted theoretical basis, particularly for observational studies for which study results are typically quite variable. Our primary and secondary outcomes were random effects.”

15.  Figure 1 suggest that it would be helpful to include boxes on the in-text study inclusion figure (for both GI and GL) in the main body of the paper. Although I understand it has been included as a supplemental figure, I think it would be helpful to show the starting number of studies with a categorization of why they were not included, instead of just starting the figure at 24 reports. The sentence about prior to 1997 is not clear.

We moved Figure S1 to be Figure 1, removing overlapping material. Figure numbers were updated as were numbers in figure titles, in-text, in Tables footnotes or columns in both the main article and the Supplemental Materials file.

16.  Line 465, “Sensitive” should be Sensitivity

Indeed, sensitivity was meant and now used.

17.  Line 470, what does “within” mean at the end of this sentence?

Thank you, we removed the spuriously appearing word “within”

18.  Line 475, “ask” should be asked; Line 495, “instant” should be instance; Line 513, do you mean “follow” instead of “fall”?

Thank you ”ask” was corrected to “asked”

Thank you ”instant” was corrected to “instance”

Thank you. To avoid confusion we replaced “fall” with decrease

19.   Line 775, I think “without statically inconsistent” should be “without being statistically significant”; Line 777, “publish” should be

a) “without inconsistency” was meant , now corrected

b) Indeed: “published” was ment., now corrected.

20.  Line 839, suggest explaining more the statement that “two studies included in the analysis in [15] have been questioned as includable, [11] others have used them similarly.”

We inserted the explanation:  “to avoid concern over possible over representation of results from the NHS I”

21.  Line 843, “sensitive” should be sensitivity; Line 934, “assumption” should be assumptions; Line 969, “inatake” should be intake; Line 980 “adjusmnet” should be adjustment; Line 987, “instrument” should be instrument

22.  Line 1003, I think the sentence “Some effects nevertheless was not excludable and was plausible considering the significance for GI” needs to be reworded

Indeed, thank you. The short paragraph was reworded as follows: “Adjustments for CORR alone was able to reduce inconsistency and heterogeneity to near zero (Table 8 row 2 for GL, n=15 studies); this also when including studies with a low validity correlation for carbohydrate (Table 8 row 1 for GL, n=21 studies). Compared with the covariate CORR, FHD was not the dominant covariate affecting the size of the T2D-GL risk relation. By contrast, when adjusting the T2D-GI risk relation for CORR, FHD was a significance factor (Section 3.2.13).

23.  Lines 1010 and 1019, the use of the phrase “at present” is confusing; Line 1011, it is not clear what is meant by “Recall adjustments”

a) “at present” was replaced by “estimated  here”

b) “at present was estimated” is now replaced by  “ was estimated here”

You might prefer “by us” instead of  “here”

c) A at Line 1011 we clarified  using the following sentence: “Salmeron et al (1997) in women [69] (NHS I at 6-y follow up) had dose-response T2D-GL risk relation estimated here to be 1.65 (1.18-2.29) per 80 g/d GL in 2000 kcal (8400 kJ) diet without adjustments for the covariates SEX, CORR, ETH and FUY, and 1.49 (1.07-2.07) after making  adjustments for SEX (equal numbers of males and females), CORR (centered on 0.7), ETH (centered on equal numbers of European-American and non-European--American ethnicities) and FUY (centered on 10-y follow up). ”

24.  Line 1042, I think here instead of T2D-GI you mean T2D-GL

Yes, indeed, thank you. This has been changed to T2D-GL

25.  Lines 1074-1075, while it is true that you included studies that had valid dietary instruments for estimating exposures, you also conducted analyses where the CORR was<0.55. Suggest mentioning that here too.

We inserted the following sentence: We also conducted meta-analyses of studies using dietary instruments that were invalid by the criterion used. Consistent with the invalidity the RR values  were not different from zero  (Tables 1 and 4).”

26.  Lines 1080 and 1088, saying that the outcome was the T2D-GI relation which was our primary outcome for GI (or for GL in the second instance) is not clear.

The two paragraphs were reworded as follow:

Among studies using valid dietary instruments the T2D-GI risk relation was 1.27 (1.15-1.40) (P<0.001, n=10 studies) which was our primary outcome for GI (Table 7 row 2 for GI). Potentially this was an underestimate because a subgroup of studies that used the most valid method of ascertainment of T2D (clinical report) showed a higher risk relation of 1.40 (1.29-1.51) (Table 7 row 3 for GI) and the risk relation remained higher for a subgroup that both used clinical report and made study-level adjustments for family history of diabetes (FDH) when RR was 1.41 (1.32-1.51) (Table 7, row 4 for GI).

3The outcome for the T2D-GL risk relation was 1.26 (1.15-1.37) (P<0.001, n=15 studies), which was our primary outcome for GL (Table 7 row 2 for GL).  Again (cf. above para for GI), this was potentially an underestimate because a subgroup of studies ascertaining T2D by clinical rather than self-report yielded a higher RR of 1.33 (1.14-1.55) (Table 7 row 3 for GL) and the risk relation remained higher in a subgroup that both used clinical report of T2D and made study-level adjustments for FHD, when the RR of 1.61 (1.18-2.12) (Table 7 row 4 for GL). 

27.  Line 1092, do you mean P<0.05 instead of P<0.50 for statistical significance?

Than you. Now P<0.05< span="">

28.  Line 1213, “ads” should be adds

Thank you. Now ‘adds”

29.  Lines 1250-1263, I think the public health relevance discussion is important. Could this be incorporated into the Abstract as well?

Thank you. We appended the following to the Abstract:.

Concerning public health relevance at the global level, our evidence indicated that GI and GL were substantial food markers predicting the development of T2D worldwide, for persons of  European ancestry and  of East Asian ancestry.

Note 1:  In addressing the requested revisions and corrections, we found the numbered paragraph in the Discussion helpful, giving some structure and marking progress of the large number of issues addressed.  If it is within the scope of the Journal, please consider keeping the Discussion paragraphs numbered.

Note 2: The following references in the above matter were already in the main article:

1.         Bhupathiraju, S. N.; Tobias, D. K.; Malik, V. S.; Pan, A.; Hruby, A.; Manson, J. E.; Willett, W. C.; Hu, F. B., Glycemic index, glycemic load, and risk of type 2 diabetes: results from 3 large US cohorts and an updated meta-analysis. Am J Clin Nutr 2014, 100, 218-232.

2.         Peto, R. Interpreting large-scale randomised evidence. John Snow lecture. Available online: http://www.youtube.com/watch?v=vybc0PsZ718&list=PL3oyPcbygtxuqmYps-aEj9Sa-ScThzMLh accessed on 25.11.2016: Oxford, 2013.
